



# Cross-Diffusion Waves as a trigger for multiscale, multiphysics Instabilities: Application to earthquakes

Klaus Regenauer-Lieb[1], Manman Hu[2], Christoph Schrank[3], Xiao Chen[1], Santiago Peña Clavijo[1], Ulrich Kelka[4], Ali Karrech[5], Oliver Gaede[3], Tomasz Blach[1], Hamid Roshan[1], Antoine B. Jacquey[6], and Piotr Szymczak[7]

[1]School of Minerals and Energy Resources Engineering, UNSW, Sydney, NSW 2052 Australia
[2]Department of Civil Engineering, The University of Hong Kong, Hong Kong
[3]Science and Engineering Faculty, Queensland University of Technology, Brisbane, QLD, 4001, Australia
[4]CSIRO, Deep Earth Imaging FSP, Kensington, Australia
[5]School of Engineering, University of Western Australia, Crawley, WA 6009, Australia
[6]Department of Civil and Environmental Engineering, Massachusetts Institute of Technology, Cambridge, MA, USA
[7]Institute of Theoretical Physics, University of Warsawa, Warszawa, Poland

**Correspondence:** Klaus Regenauer-Lieb (regenau@gmail.com)

**Abstract.** Theoretical approaches to earthquake instabilities propose shear-dominated instabilities as a source mechanism. Here we take a fresh look at the role of possible volumetric instabilities preceding a shear instability. We investigate the phenomena that may prepare earthquake instabilities using the coupling of Thermo-Hydro-Mechano-Chemical reaction-diffusion equations in a THMC diffusion matrix. We show that the off-diagonal cross-diffusivities can give rise to a new class of waves known as
cross-diffusion waves. Their unique property is that for critical conditions cross-diffusion waves can funnel wave energy into a quasi-stationary wave focus from large to small-scale. The equivalent extreme event in ocean waves and optical fibres leads to the appearance of 'rogue waves' and high energy pulses of light in lasers. In the context of hydromechanical coupling, a rogue wave would appear as a sudden fluid pressure spike on the future fault plane. This is here interpreted as a trigger for the ultimate (shear) seismic moment release.

## 1   Introduction

Part 1 (Regenauer-Lieb et al., 2020) introduced a (geo-)wave mechanics, physics-based formulation for deciphering patterns of multiphase material instabilities from the molecular scale to any larger scale. Although the paper is formulated for earth sciences the approach constitutes a generic theory for any material and therefore lacks experimental evidence. In the second part we investigate whether the approach can be applied to a real world geological system.

The challenge to come up with an approach that defines new concepts for the application of multi-scale non-equilibrium thermodynamics to earth system science was originally posed in the 'Patterns in our Planet' conference in Victor Harbor, South Australia in May 2008 (Ord et al., 2010). Patterns in our planet encode information on reaction-diffusion processes repeating themselves over multiple scales such that a magnified view of the structure looks like a copy of the structure itself (self-similarity) (Hobbs et al., 2011). The connection between these patterns as dissipative structures of reaction-diffusion

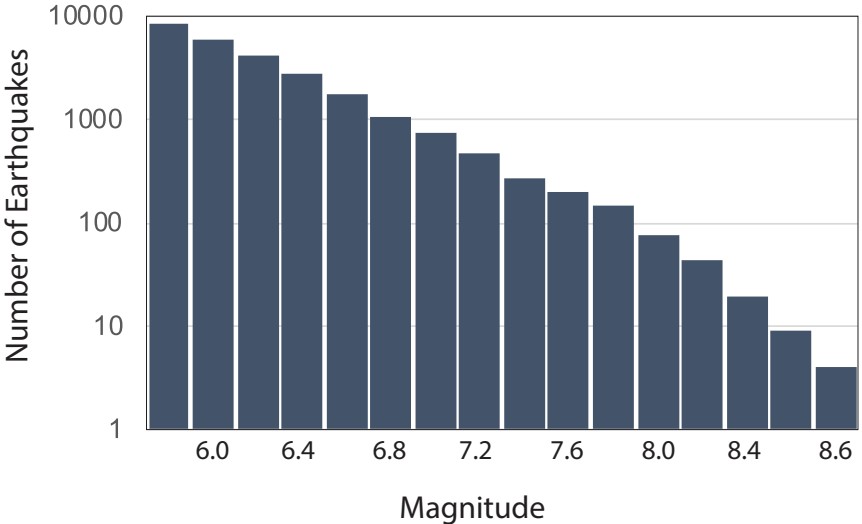

**Figure 1.** Earthquake frequency-magnitude histogram prepared from the Global Instrumental Earthquake Catalogue, Version 7.0 - released on 9/4/2020 by the ISC-GEM (Di Giacomo et al., 2018; Storchak et al., 2015, 2013). Global quakes have a fractal power law relationship of *log(Number of Earthquakes) = 10.23 + 1.06 (Magnitude)* with an $R^2 = 0.98$. The log-log relationship between numbers of earthquakes and their moment magnitude on all sizes of completeness of the catalogue suggests a simple underlying reason why the physics of the very small influences the physics of the very large in a multifractal cascade of instabilities. Sethna et al. (2001) postulates that this can be explained by the existence of a critical thermodynamics force above which an earthquake starts to slip. Accordingly, plate motions self-regulate (Sornette and Pisarenko, 2003) to be exactly at this critical point for potential failure at all scales.

systems (Ball, 2012) and their role in thermodynamic far-from-equilibrium systems was originally described by Prigogine and co-workers (Kondepudi and Prigogine, 1998). An application of self-diffusing reaction-diffusion equations to mineralising systems has been proposed recently (Oberst et al., 2018). This contribution investigates geological applications of the possible relation to the reaction-diffusion wave phenomenon due to the new addition of the cross-diffusion term proposed in part 1 (Regenauer-Lieb et al., 2020). Along the same vein, the paper also attempts to give a somewhat simpler description of the

theory from a chemical perspective putting part 1 into context with other more familiar theories as well as recent further developments thereof.

A prime example of how the physics of the very small appears to influence the physics of the very large is the earthquake instability (Sornette, 1999; Crampin and Gao, 2015). We therefore use earthquakes as a topic to discuss the roots of self-similarity underpinning the log-log frequency magnitude relationship and many other similar relationships in nature shown in

Fig. 1. To simplify the equations and address a frequently overlooked deformation mode we use the Helmholtz decomposition presented in part 1 (Regenauer-Lieb et al., 2020) and only discuss examples for dissipative pressure ($P$) waves. The rich dynamics of superposing dissipative pressure ($P$)- and shear ($S$)- waves will be subject of future research.



Working Hypothesis: This paper develops the hypothesis that patterns in our planet encode information of dissipative structures in the form of standing waves that can appear when a cross-diffusion term is added to reaction-diffusion equations
(Berenstein and Beta, 2012). We develop a continuation of a solid mechanical cross-diffusion formulation in which slow damage waves (Hu et al., 2019) have been identified as a precursor phenomenon to macroscopic failure of porous materials under load. The failure pattern induced by these waves imprints a characteristic bar-code like signature around the main failure zone. If it is possible to decipher these patterns as frozen-in stationary states of reaction-diffusion systems, the approach may provide new avenues to investigate one of the most difficult unsolved problems in earth sciences through direct geological observations.

Complex earth/material dynamic reaction-diffusion processes occur under extreme temperature and pressure conditions and on time-scales inaccessible to the direct human observer (Kohlstedt and Holtzman, 2009). Therefore, reliable empirical experimental engineering approaches for estimating the risk of failure of Earth materials under both engineering and plate tectonic loads are missing (Grigoli et al., 2018). A particular challenge is to replicate nature's pattern forming, self-organised critical conditions (Fig. 1) at micro-scale in the lab, as the earth's long-range feedbacks are missing.

We use a thermodynamic-based continuum mechanics approach where we consider conservation of mass, conservation of linear momentum, conservation of angular momentum, conservation of energy, and the second law of thermodynamics as the basic set of coupled partial differential equations (*pde*). These equations form the basis of deterministic continuum mechanics while the strong form of the second law is used as a bridge between statistical mechanics and continuum approaches (Ostoja-Starzewski, 2008). In this respect the thermal (T) *pde* has a special role as it is tied to the entropy evolution and thereby 50 encapsulates the uncertainty quantification for time-dependent processes through the fluctuation theorem (Evans and Searle, 2002). For a complete approach, we must look for a mathematical description that comprises all four coupled partial differential equations in a holistic way including the important uncertainty quantification.

The wave-mechanics approach offers just this opportunity as it encompasses all conservation laws including the uncertainty relationship. The wave mechanics approach of part 1 (Regenauer-Lieb et al., 2020) offers in particular a fundamental physics-55 based first guess to investigate these critical domains that are not directly accessible. We thereby identify critical conditions for many important earth science problems such as the physics of earthquakes, extraction of geothermal energy, safety of nuclear waste disposal, reservoir engineering for oil and gas, the formation of mineral deposits, induced seismicity, natural hazards, groundwater management, and $CO_2$-sequestration and utilization. In sections to come, we will illustrate the potential first application of the theory and depict its relation to other similar approaches in different disciplines.

## 60   2   THMC length and time scales

The decoupling of dynamic processes on vastly different length and time scales is one of the most powerful methods of upscaling techniques. The objective, thereby, is to reduce the level of complexity by coarse-graining (Sethna, 2006; Hanasoge et al., 2017) a complex thermodynamic system to a larger system such that the dynamics of the lower scale can be homogenized into an effective material property for the larger scale (see Fig. 2). The dynamical system maps onto itself in a self-similar 65 fashion so that the dynamic behaviour under external loads is the same irrespective of the scale considered. This so-called





coarse-graining approach leads to time-evolution of the thermodynamically averaged larger systems that have significantly simpler time-evolution equations. The dynamic processes at lower scale are assumed to have relaxed to a quasi-steady state (local thermodynamic equilibrium) and contribute to the larger system through what is described in statistical physics by an order parameter characterising the scale-invariant free energy of the statistical volume considered (Sethna, 1992). This situation

can lead to a cascade-like hierarchy of singularities described by a series of local power laws.

     The system would then be expected to feature a multiscale combination of local power laws of Thermo-Hydro-Mechano-Chemical (THMC) reaction-diffusion equations leading to multifractal distributions where different scales have different fractal properties (Stanley and Meakin, 1988). This multifractality relies on a random multiplicative process of each underlying physics and a coupling of the critical point phenomena into the universality relationships of the THMC-coupled processes.

The combination of a thermally activated rupture with a long memory stress relaxation was proposed as a possible mechanism to explain the multifractal scaling of the Californian earthquake catalogue (Sornette and Ouillon, 2005). The multifractality hypothesis of the Californian dataset has been reinvestigated in a more recent multiscale analysis of the micro-, meso- and macro-scale subsequences showing that the macro-scale spectrum indeed has the strongest multifractality of the three scales, thus strongly supporting the hierarchy of scales (Fan and Lin, 2017). This finding calls for a true multiscale formulation for

earthquake physics, where the largest scale geodynamic driver is coupled to the smallest scale singularity in a multiphysics hierarchical cascade of instabilities.

     When extrapolating the findings from the Californian dataset to the global earthquake dataset we infer that the multifractal, so-called singularity spectrum (Turiel et al., 2006) contains most -but unfortunately not all- information about the physics of earthquakes (see Fig. 1). The continuous spectrum of multiscale exponents contained in the global singularity spectrum sug-

gests that there is a mechanism that is capable of coupling the various THMC reaction-diffusion equations. Individual coupling mechanisms have been discussed such as a creep activation mechanism (Sornette and Ouillon, 2005), shear heating (Ogawa, 1987; Regenauer-Lieb and Yuen, 1998; Braeck and Podladchikov, 2007), thermally induced fluid pressurization (Vardoulakis, 2001; Rice, 2006) and a mixed process between frictional slip failure and the shear fracture of intact rock (Ohnaka, 2003). A generic physics-based formulation for investigating multifractality of the earthquake mechanism that does not single out

individual processes is still lacking. The current work attempts to develop such a generic framework.

     A possible candidate for cross-scale communication, could be the propagation of cross-diffusional waves which can tie several or all THMC reaction-processes together in a convolution operation (Regenauer-Lieb et al., 2020). The time-domain convolution operation of THMC waves can be seen in the frequency-domain as a filter for the dominant earthquake coupling mechanism, sharpening or smoothing certain cross-diffusion waves. In this sense, the earthquake physics problem may,

therefore, be condensed to the problem of how to couple instabilities across scale such that a dominant wave can lead to a self-similar macroscale instability mechanism which we propose to be an extreme form of a sharpening filter in the language of signal processing. In the physics field, this phenomenon is known as a rogue-wave (Eberhard et al., 2017b). To verify or falsify the rogue-wave hypothesis as a potential earthquake trigger mechanism, we need to discuss the interplay of the vastly different time- and length- scales of the THMC reaction-diffusion processes.




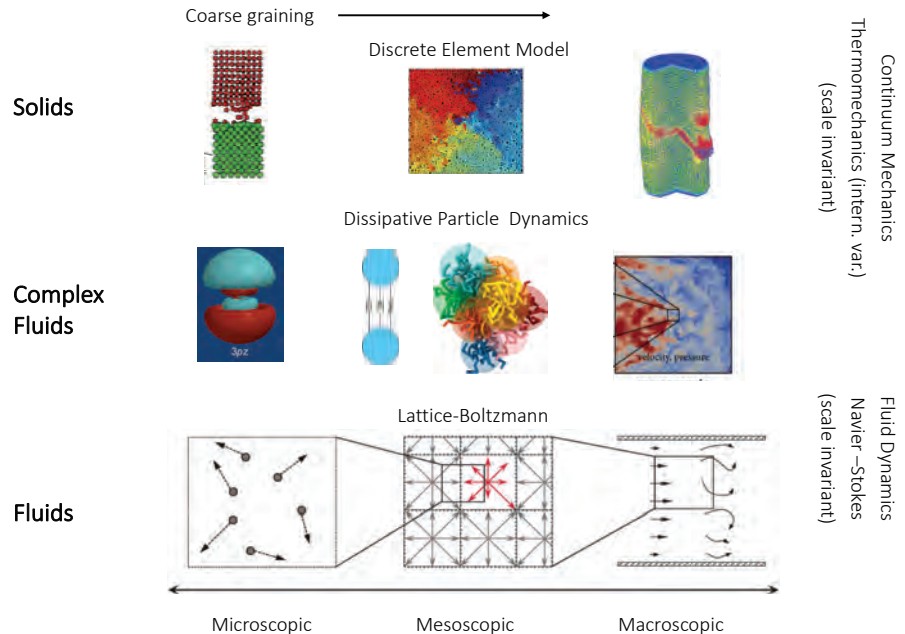

**Figure 2.** Coarse graining approaches in continuum and fluid dynamics. The microscale has the richest information and highest degrees of freedom. Coarse graining reduces the complex dynamics to time-evolution equations with a less detailed level of description. In the continuum scale only reduced variables are retained. In the case of the Navier-Stokes equation for fluid dynamics, these are pressure and velocity. The wave mechanics approach presented here is a meso-scale approach that opens the path for an analytical reduction of variables rather than a computational coarse-graining method based on averages. This enables the derivation of physics-based macro-scale constitutive equations for the continuum scale. This approach is complementary to the numerical coarse-graining techniques shown in this figure.

The characteristic time and length scales of the thermodynamic THMC processes can be evaluated by calculating the relaxation time of a random perturbation from the THMC reaction term $R_i$. Characteristic time- and length- scales emerge from quasi-steady-state (time-independent) solutions of the diffusive (relaxation) response of the reaction-diffusion system characterised by the equation

$$\frac{\partial C_i}{\partial t} = \nabla \cdot (\zeta_i' \nabla C_i) + R_i,\tag{1}$$

where $C_i$ stands for the diffusing species/processes (e.g. temperature, fluid or mechanical pressure, chemical species) and $\zeta'$ for the respective diffusivity with the subscript $i$ denoting the THMC processes. Linearising the diffusion equation to first order we obtain

$$\frac{\partial C_i}{\partial t} = \zeta_i \nabla^2 C_i + R_i.\tag{2}$$





The time scale or the relaxation process is now entirely characterised by the self-diffusion constants $\zeta_{T,H,M,C}$ and the reaction rate constants $R_i$.

## 2.1 Self-Diffusion length/time scale

For a discussion of the length scale encountered through the diffusion of a local reaction source term of the THMC process (i.e. local heat release, fluid or mechanical pressure source term, chemical reaction) we treat the concentration $C_0$ released by the reaction at $t_0$ as an initial condition and set $R_i = 0$ for the remainder of the process $t > 0$. We use this instantaneous source as a point source in an infinite plane. This linear partial differential equation requires one initial condition (local release of concentration $C_0$ as a point source due to the reaction at $x = 0$) and two boundary conditions for solution. The boundary conditions are $C_i = 0$ at plus and minus infinity. Mass (momentum, energy and entropy) balance requires that at any time the concentration (mechanical/fluid pressure, heat content) is:

$$C_0 = \int_{-\infty}^{\infty} C_i dx. \tag{3}$$

If we assume that the initial and boundary conditions are unchanged by a scale change, as required by the coarse graining assumption, the well-known solution method is obtained by scaling the equation with a characteristic diffusional length scale. This diffusion length scale defines the characteristic length scale of propagation of the information from a local perturbation, e.g. for chemistry an instantaneous point source in the concentration of the diffusing species being released at the origin $x = 0$ at $t = 0$ in a given time $t_d$. For fluid flow and mechanics this would be a local source of fluid or mechanical pressure, and for temperature this perturbation would be a local heat source. Any scale that is significantly larger than the diffusional length scale is considered to be unaffected by the diffusion front for a given diffusion time $t_d$. We will later on reinterpret the diffusion length as an uncertainty measure. The characteristic diffusion length scale is:

$$L_d = \sqrt{4\zeta_i t_d}. \tag{4}$$

which is used as part of the scaling method for solving Eq. 2 described in standard textbooks e.g. (Crank, 1975). Using the previously described initial and boundary conditions, the diffusion front spreads radially in a concentration profile

$$C_i(x,t) = \frac{C_0}{L_d\sqrt{\pi}} \exp-\left(\frac{x}{L_d}\right)^2. \tag{5}$$

The solution shown in Fig. 3 illustrates the fast decrease of the amplitude with increasing time coupled with an increase in $L_d$. The normalisation over $\frac{1}{L_d\sqrt{\pi}}$ in Eq. 5 ensures that the area under the curve always remains the same and satisfies Eq. 3 thus defining a Gaussian scale space. The Gaussian probability density function (here called a Gaussian wavelet) is often used in probability theory where the diffusion length is related to the standard deviation, its square to the variance and the function centroid to the mean.





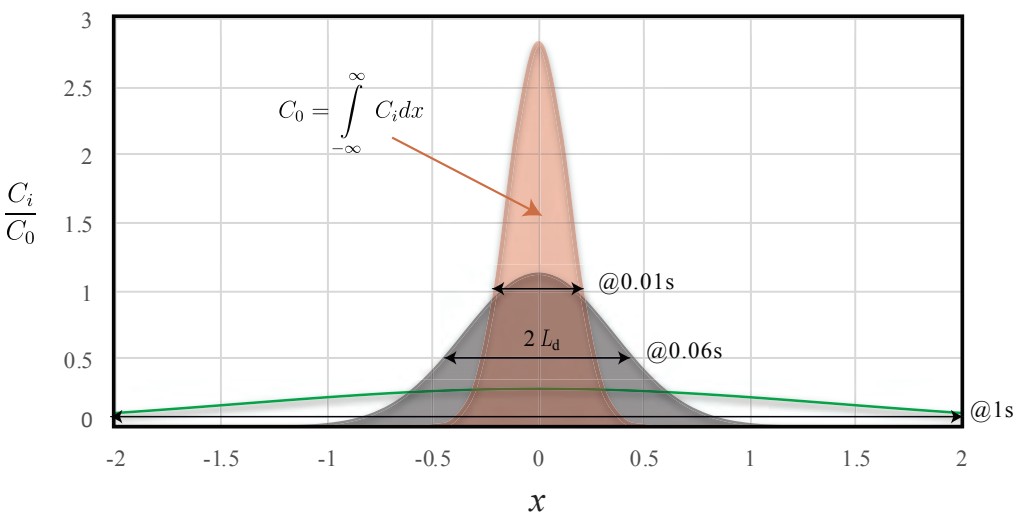

**Figure 3.** Illustration of the diffusion length sale using the example of a point source pulse of concentration applied as an initial condition in a plane at $t = 0$ with $C_0 = 1$. An arbitrary diffusivity of $1\ m^2 s^{-1}$ is selected. Three time snapshots are shown at (0.01, 0.06, and 1 s) showing that the normalised concentration reduces while the pulse broadens as a Gaussian wavelet (Gaussian kernel in the language of image processing) with a wavelength characterised by the diffusion length scale $L_d$. The broadening of the wavelength is defined by twice the diffusion length $L_d$ due to the bi-directional diffusion process. The diffusion length scale describes the wavelength at exactly the same fraction (here $\frac{1}{e} \approx 0.37$) of the maximum magnitude of concentration at a given time step.

| Self Diffusion | Diffusivity $[m^2/s]$ | Process time $[s]$ | Diffusion Length $[m]$ |
|---|---|---|---|
| $\zeta_T$ | $10^{-6}$ | $10^{12}$ | $10^3$ |
| $\zeta_H$ | $10^{-5} - 10^{-1}$ | $10^2$ | $6 \times 10^{-2} - 6 \times 10^0$ |
| $\zeta_M$ | all | all | all |
| $\zeta_C$ | $10^{-19} - 10^{-15}$ | $10^{12}$ | $6 \times 10^{-4} - 6 \times 10^{-2}$ |

**Table 1.**

Typical diffusion length scales for geological THMC processes (Regenauer-Lieb et al., 2013a). Note, the mechanical diffusion length scale can range over all scales. As an example the visco-elastic diffusion length scale for elasto-dynamic earthquake events is very short. However, prior to the catastrophic event stress diffusion can range over all scales and can couple all processes through elasto-visco-plastic creep processes.



## 2.2 Reaction-Diffusion length/time scales

So far, we have discussed a point source reaction as an initial condition at time $t_0$ and $x = 0$ with no additional time-scale other than the self-diffusion process afterwards. This leads to a decaying and broadening diffusion wavelet (Fig. 3). When considering an active source term, the solution turns into a propagating wavefront which was first discovered in 1906 by Robert Luther. An English translation of the original article appeared in the Journal of Chemical Education (Luther, 1987). The same phenomenon was rediscovered 30 years later and is now known as Fisher or Fisher-Kolmogorov equation (Adomian, 1995). Fisher originally discussed the reaction-diffusion equation to calculate the propagation of a mutant virus in an infinite domain (Fisher, 1937). Following Showalter and Tyson (1987) we recast Fisher's arguments for calculating the minimum speed of the propagation of the mutant gene at long time scales into a discussion of a propagation of a chemical reaction front. For the example discussed in Fig. 3 consider the following autocatalytic chemical reaction:

$$[A] + n[C] \longrightarrow (n+1)[C],$$

(6)

where $A$ is the reactant and $C$ is the catalyst, $n$ is the autocatalysis order with the number of reaction steps with $n = 1$ being the elementary reaction (Valero and Moyano, 2017). The square bracket indicates the number of moles. The reaction rate is $R_i = k[C] = \frac{d[[C]}{dt}$ with $k$ being the first-order reaction-rate here defined with respect to the reaction product rather than the reactant, hence the positive sign. This reaction depends on the concentration of only one reactant. The system may have other reactants, but these are not influencing the rate of the reaction. In chemistry this situation is known as a first-order reaction. As we are interested in a moving reaction front, we may consider just the initial response of the reaction. Thus instead of the typical sigmoid solution, which considers the depletion of [A] in the course of the reaction, we may consider a non-physical infinite pool, where [A] is not depleted by the reaction, and integrate the first-order reaction by an exponential growth of the reaction product, with

$$[C(t)] = [C_0] \exp(kt).$$

(7)

The minimum velocity of the propagating reaction-diffusion wavelet that is triggered by an initial delta-function reaction at time $t_0$ can be evaluated by extending the solution in Eq. 5 to consider the exponential reaction product and obtain:

$$[C(x,t)] = \frac{[C_0] \exp(kt)}{L_d \sqrt{\pi}} \exp - \left( \frac{x}{L_d} \right)^2 .$$

(8)

Separating variables,

$$[dC] = \frac{\partial [C]}{dx} + \frac{\partial [C]}{dt},$$

(9)

and substituting the separation into Eq. 8 we obtain an ordinary differential equation (*ode*)

$$\frac{dx}{dt} = \frac{2k\zeta t}{x} + \frac{x}{2t} - \frac{\zeta}{x},$$

(10)





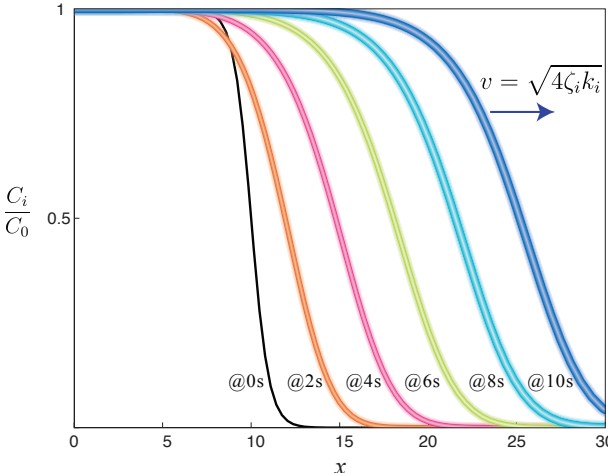

**Figure 4.** Illustration of the propagating wavefront of the Fisher-Kolmogorov equation which is Eq. 6 with a nonlinear source term $R_i = k_i C_i (1 - C_i)$. Kolmogorov et al. (1937) showed that any initial concentration vanishes for large $x$ and evolves to a travelling wave solution with a minimal velocity $v = \sqrt{4\zeta_i k_i}$. The reaction-diffusion exhibits a bistable equilibrium. One stable equilibrium is achieved when the initial concentration is below the activation of a wave and the system is resting and the other is the travelling-wave solution with a minimal velocity $v$. We show here the non-dimensional solution for the example of a $tanh$ initial condition (black curve) which quickly converges to a travelling waveform with a non-dimensional speed of 2. Various solution techniques exist. A convenient method is to turn the partial differential equation (Eq. 2) into an ordinary differential equation by using Chebyshev polynomials (Towers and Jovanoski, 2008).

where $\zeta$ is the diffusivity. This shows that for large $t$ and large $x$ the last two terms of Eq. 10 vanish, and a constant wavefront velocity is expected to approach

$$v = \sqrt{4\zeta k}, \tag{11}$$

which is a lower bound of the steady front velocity at large $t$. We can use this speed as a quasi-steady-state solution of the composition wave triggered by the reaction and obtain a quasi-steady state diffusion length scale of $L_d = \frac{\zeta}{v}$ that propagates

with the wavefront. This diffusion front is the only information, that is carried by the wave at long length and time scale, and highlights one of the fundamental differences between reaction-diffusion waves and elastic waves. Reaction-diffusion waves quickly diffuse information of the initial condition and at long time scales are only characterised by the competition between reaction rates and diffusion rates. Pure elastic waves, when propagating without damping, diffraction and scattering, carry the full source information through their entire travel path.

The above example of a chemical autocatalytic reaction illustrates the key features of reaction-diffusion equations in many scientific disciplines and applies to all $C_i$'s and $R_i$'s of THMC systems. Adding an active non-linear source term (e.g. $R_i = k_i C_i (1 - C_i)$ for the Fisher-Kolmogorov equation) into the diffusion equation can lead to the interesting phenomenon of the





generation of a self-oscillatory excitation wave, where after Fisher's work on the topic progress was mainly made in the Russian literature triggered by the seminal work of Kolmogorov et al. (1937) discussed in Fig.4.

In Russian literature, the term "autowave" was introduced (Ostrovskiǐ, 2015). The autowave phenomenon is well-studied for understanding the electrical nerve impulse that leads to contraction of the heart muscle (Antonioletti et al., 2017). However, it constitutes a fundamental class of waves encountered in all reaction-diffusion systems in physics, biology, and chemistry (Vasil'ev, 1979). The principal difference to classical wave equations that are based on hyperbolic differential equations, is that the autowave phenomenon arises from a non-linear source term in parabolic equations. When adding a cross-diffusion term

autowaves can turn into standing waves (Berenstein and Beta, 2012) having completely different properties to classical standing wave solitons. This difference defines a new class of dissipative waves which, according to Tsyganov and Biktashev (2014), presents an entirely different "world" to the waves encountered in integrable conservative systems such as the elastic-wave phenomenon. This phenomenon will be discussed further in the section on cross-diffusion waveforms where we will discuss the most important property for nucleation of earthquakes. This is the potential of multiscale funnelling of wave energy from

the environment into a localised standing wave. Before going there, we continue with a discussion of the reaction-diffusion equation without the cross-diffusion term in order to systematically describe its basic characteristics.

The reaction-diffusion time-scales of a single reaction-diffusion equation can be expressed by a generalization of the Damköhler number for chemical reaction-diffusion processes to all THMC couplings. The Damköhler number describes the ratio of the diffusion over the reaction time or the equivalent ratio of the reaction over the considered diffusion rate.

$$\mathrm{Da_i} = R_i t_d = \frac{R_i L_d{}^2}{4\zeta_i}.$$   (12)

This ratio now defines a new time-scale and replaces the purely diffusive time-scale discussed in the spreading wavelet shown in Fig. 3 with a propagating wavefront solution that self-supports its shape. This is illustrated here only for an infinite autocatalytic source term in the reaction-diffusion system (Fig. 4). For a finite autocatlytic reaction the reactive source term has a growth function which is sigmoid, also called a logistic function. This means that the reactant initially grows exponentially, similar to

the infinite source, followed by linear reactant growth and a final zero growth branch. These general solutions imply different shapes of the self-supporting propagating wave and, depending on the value of the diffusivities, also a finite life-time of the wave.

This geologically more relevant situation is described in Molotkov and Vakulenko (1993). The authors describe generalised reaction-diffusion systems and find that the wave behaviour depends on only three parameters. For small wavefront curvature

the autowave is described by the two parameters of the infinite source term solution discussed above, while for the more general case the normal velocity of the front may contain the front curvature as an additional parameter. The additional dependence of the wave function on the curvature of the concentration field arises because of the fact that $\frac{\partial C_i}{\partial t}$ is proportional to the curvature of the wavefront. For regions of the wavefront where the curvature is negative the concentration must decrease at a rate proportional to the magnitude of the curvature. Conversely, the concentration must increase where the wavefront curvature

is positive.





Summarizing the above findings, we can now characterise the reaction-diffusion thermodynamic system by five key features: i.) a bistable or multistable (for several reactions) region with a stable stationary mode and a mode for the nucleation of propagating autowaves above a critical activation threshold; ii) in the activated state, the wavefront separates two regions, a local region characterised by the particular THMC diffusional length scale $L_d$ affected by the reactions $R_i$, and a large region

at $> L_d$ which is outside the reaction-diffusion wave; iii) for long time scales, the wavefield is governed by characteristic self-oscillatory motions which for bistable systems are described by just three parameters. For multistable systems chaotic oscillations are expected (Molotkov and Vakulenko, 1993). For the analysis of this complicated system we will propose to use perturbation theory and illustrate the approach through some basic concepts of signal processing. In the bistable system the Fisher-Kolmogorov wave is a self-propagation dissipative wave at a characteristic wave speed whose lower limit can be

quantified by the square root of the Damköhler number times the diffusivity normalised by the characteristic diffusion length scale $L_d$; (iv) the propagating wave exponentially scatters information about its initial condition, and the wavefield only carries information about the dissipative properties into the far-field; this is an important differentiation to waves in the conservative system (e.g. elastic waves) where the wave at a long-distance still carries information about its initial conditions; (v) the wave speed thus becomes a fundamental material constant defined by the rates of the dissipative THMC processes as:

$$v_i = 4 \frac{\zeta}{L_d} \sqrt{\mathrm{Da_i}}. \qquad (13)$$

As the spreading wavefronts are self-supporting and can propagate upwards in scale, we propose that this material velocity not only applies to the above discussed chemical reaction-diffusion equations but to all reaction-diffusion equations of the THMC-coupled system. In this proposition, the propagating multiscale and multiphysics waves provide the capacity to link the different THMC-length-scales and could explain the multifractal nature of earthquakes. The approach allows a significant simplification

of the earthquake physics problem as the exponential rate of approaching the Kolmogorov limit of a self-oscillating wavefront shown in Fig. 4 can be simplified by replacing the nonlinear source term through a set of at least two linear coupled partial differential THMC diffusion equations which will be discussed in the next section. At a given scale, the Kolmogorov wave velocity limit allows a characterisation of the important physics in terms of the wave velocity. With the autowave approach, one can turn any non-linear perturbation of the local source into a linear propagating waveform only governed by the dissipative

material properties. Autowaves will by themselves recover a characteristic wavefield dictated by the reaction-diffusion rate constants.

This characteristic behaviour is used, for instance, in medicine where autowaves are encountered in many fields. The electrical nerve impulses that drive a regular heart beat are an example (Antonioletti et al., 2017). The authors describe how the characteristic recovery of the autowave waveform after a random perturbation can be used, for instance, for defibrillation strate-

gies (a small electrical stimulus applied through a pacemaker) for treatment of life-threatening heart arrhytmia. In this sense, earthquake physics might profit from an understanding of the partial differential equations developed in mathematical biology, epidimiology (wave-like propagation of viruses) and other biomedical applications for which numerical open-source tools are available. One such tool is the "heart beat box" (Antonioletti et al., 2017) which uses a mathematical formulation of the human





heart in terms of a coupled electro-mechanical reaction-diffusion equations similar to the coupled reaction-diffusion equations
discussed above.

The foregoing results were based on the generalisation of a first-order autocatalytic reaction. Very few practical examples will
be first order, and we need to consider the generalised case of a reaction that depends on the concentrations of a second-order
reactant or any number of additional reactions and their equivalent THMC processes. To extend the discussion to more than one
reaction, we apply the Fisher reaction term to more than one interdependent reactant. That means in chemical terms that the
concentration of second- or higher-order reactions depends on the concentration of a second- or higher number of species. This
will add additional dynamics to the system response. To isolate the effect of this additional feedback it is useful to neglect the
spatial response and neglect diffusion. Mathematically, such an interdependent coupling of reaction rates is expressed by just
considering the reaction term for two cross-linked autocatalytic equations. This is known as the Lotka-Volterra predator-prey
model (Lotka, 1920), which beautifully illustrates the generic behaviour of a coupled system in time. The interaction with the
spatial response will be discussed later.

### 2.3 Periodicity in time: Lotka-Volterra Waves

The Lotka-Volterra predator-prey model couples two autocatalytic reactions

$$
\begin{aligned}
A + X &\rightarrow 2X \quad \text{prey } (X) \text{ have an infinite supply of food } (A) \text{ and multiply} \\
Y + X &\rightarrow 2Y \quad \text{predators } (Y) \text{ eat prey } (X) \text{ and multiply as a consequence}\,.
\end{aligned}
\tag{14}
$$

Coupling between the two autocatalytic reactions occurs because the predators can only multiply if they eat prey. Similar to
the above assumption for the derivation of the limiting wave speed, we assume that the food supply $A$ for the prey is infinite
and without predators there would be an exponential growth of prey. In chemical terms for the molar concentrations, this leads
to the following rate equations (Lotka, 1920).

$$
\begin{aligned}
\frac{d[X]}{dt} &= k_1[A][X] - k_2[X][Y] \\
\frac{d[Y]}{dt} &= k_2[X][Y] - k_3[Y]\,,
\end{aligned}
\tag{15}
$$

where $k_1$ and $k_2$ are reaction rates of the autocatalytic reactions derived from empirical/phenomenological postulates. To
close the system a third rate $k_3$ must be introduced that quantifies the rate of death of predators as the death rate of prey is
already included in the reproduction equation for the predator. We can evaluate the equilibrium points of the system by setting
$\frac{d[X]}{dt} = \frac{d[Y]}{dt} = 0$ and obtain following condition for the rates:

$$
\begin{aligned}
X &= \frac{k_1}{k_2}[A] \\
Y &= \frac{k_2}{k_3}\,.
\end{aligned}
\tag{16}
$$

Another equilibrium point is where the predators consume all prey leading to the extinction of both species. Integrating the
Lotka-Volterra *ode* numerically reveals the fundamental oscillatory behaviour of coupled reactive systems shown in Fig. 5.

The Lotka-Volterra oscillator applies to second-order and higher-order reactions where multiple oscillators may be encoun-
tered. We posit here that the fundamental analysis can be transferred from the classical biological system (predator-prey,



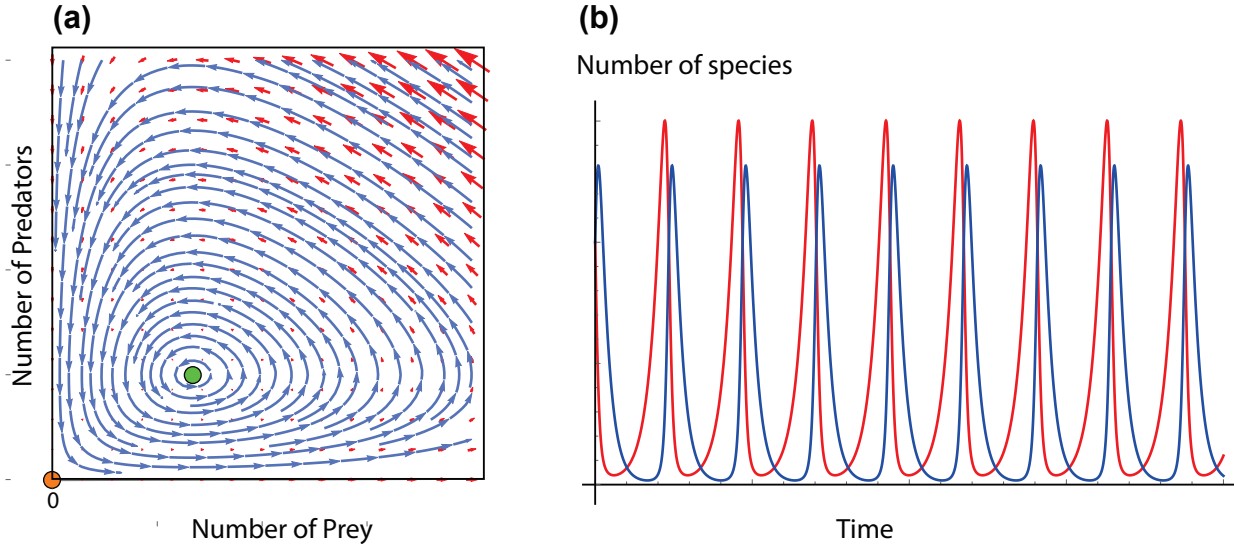

**Figure 5.** The Lotka-Volterra oscillator represents the fundamental behaviour of coupled reactive systems. (a) The phase plot with the two equilibrium points (green and orange dots). (b) The corresponding time-evolution of one of the cycles in the phase plot. Predators are in red and prey in blue. The figure shows a simple calculation of the Lotka-Vollterra oscillator model in Mathematica (2020). Lotka-Volterra Tutorials are also available allowing for incorporation of a fourth rate constant for the level of predation (Mathematica, 2020).)

infectious diseases, etc.) and chemical reactions to all coupled reactive source terms of the THMC reaction-diffusion system. When looking for a possible Lotka-Volterra oscillator in earthquake dynamics, one would start with trying to identify first a perfectly regular oscillator which would involve only two coupled THMC equations. To then generalise the approach one would continue to the generalised Lotka-Volterra oscillator, such as the three-species oscillators, which explain the transition to chaos. An example is shown in Fig. 12.4 in Flake (1998).

### 2.3.1 Lotka-Volterra type earthquake sequences

There are examples of characteristic earthquakes with some regularity in their recurrence, e.g. the Parkfield example (Wiemer and Wyss, 1997)), and also the Episodic Tremor and Slip (ETS) events recorded in Japan, Cascadia and Hikurangi subduction system (Gomberg, 2010). These have been interpreted with a perfectly periodic oscillator model (Poulet et al., 2014b) shown in Fig. 6. It was shown that a coupling of the temperature reaction-diffusion equation with the pressure reaction-diffusion equation (Alevizos et al., 2014; Veveakis et al., 2014) can lead to the characteristic perfectly periodic Lotka-Volterra-type oscillatory response in the temperature-pressure plot (Fig. 6). The authors also show an example where, by considering an additional oscillator, a transition to chaos can be modelled (Poulet et al., 2014b). The basic element of the model is a chemical decomposition function of the type $AB \rightleftarrows A + B$. This dissolution decomposition reaction is different to the autocatalytic reaction described above. The equivalent autocatlytic element in this decomposition reaction is introduced through shear heating feedback during





**(a)**

**(b)**

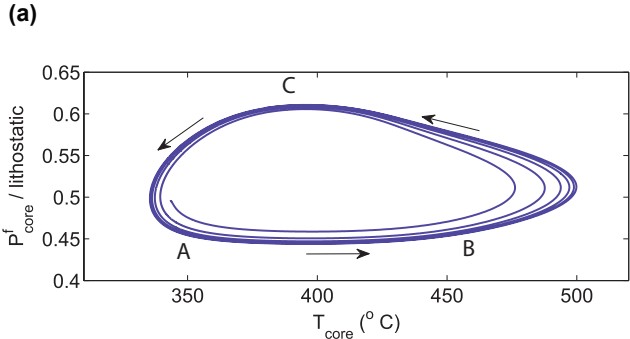

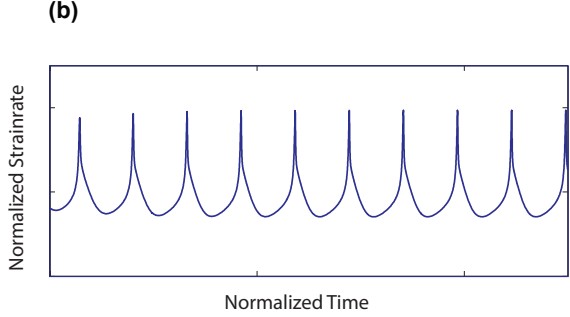

**Figure 6.** The characteristic period of the Cascadia ETS sequence interpreted with a reaction-diffusion oscillator model (Poulet et al., 2014b). (a) The phase diagram between $A$ and $B$ corresponds to the slow creep of the serpentinite slowly raising the temperature to the critical level for the onset of the dehydration reaction. The segment between $B$ and $C$ corresponds to the dehydration reaction that coincides with the tremor and slips event. The following segment between $C$ and $A$ is the diffusion-dominated field. (b) shows the corresponding normalised strain rate with the sharp peak corresponding to the tremor event.

fault slip, triggering thermally induced dissolution. Poulet et al. (2014b) numerically investigated the potential candidate source mechanism of a thermally induced dissolution reaction of serpentinite in the fault zone originally proposed by Obara (2002).

In that model solid serpentinite (phase $AB$) decomposes into a solid phase $A$ (Antigorite) and a fluid phase $B$ (water) upon application of a heat source from shear heating.

## 2.4    Periodicity in space: Cross-diffusion Waves

It is encouraging that a simple chemical reaction model of the normally fine-grained serpentine crystal at say millimetre scale can be used to model an ETS instability at plate scale, say around 100-1000 km scale. Communication of information over eight

to nine orders of magnitude may hence be possible under special circumstances. This leads to the proposition that progress can be made by investigating the multiscale physics of the THMC reaction-diffusion system in the Earth in further detail. In addition to the timing and displacement information obtained from GPS and seismic stations, the spatial information related to the diffusion term could be investigated to verify the model. A key observable would be the propagating wavefront that would be expected from the above discussion on reaction-diffusion time scales.

In the case of the serpentinite decomposition reaction, the diffusive wavefront would propagate normal to the main fault plane as the chemical decomposition reaction involves significant volume reduction which would lead to a contraction of the central fault plane and a characteristic width of the fault plane. This process zone is not exposed in active subduction systems but can be found in geological exposures of fossil faults. An excellent example is the Glarus Thrust which has been modelled by the Lotka-Volterra-type oscillator approach using a carbonate decomposition reaction (Poulet et al., 2014a). The model includes

diffusion and therefore the expected spatial response can be tested as well. The model is shown to be capable to reproduce the process zone around the central fault plane which is documented in many fault zones (Chester et al., 2013). Another textbook





example is the Punchball Fault in California where a series of deformation mechanisms were described (Schulz and Evans, 2000).

Synchronous or staggered deformation mechanisms are a general observation (Vermilye and Scholz, 1998) and require expansion of the chemical decomposition reaction model to include other important meso- to macroscopic deformation mechanisms. Additionally, microstructure, geometric complexity, and multiple deformation mechanisms lead to a stochastic element at the meso-scale whose role in the transfer of the information from micro- to large-scale in real geological applications is not yet discussed. In the following we will discuss whether these meso-scale stochastic processes lead to a loss or an enhanced coupling from small to large scale.

The problem can be addressed by a statistical mechanics approach to the coarse-graining problem. In Fig. 2 we proposed that the missing element for considering mesoscopic complexity in a physics-based earthquake model is a meso-scale approach that captures the link between the vast differences in the diffusional length scale of THMC processes shown in Table 1. In part 1 (Regenauer-Lieb et al., 2020) we proposed that a meso-scale approach can be developed by decomposing the large-scale reaction term $R_i$ into a meso-scale reaction term which requires a meso-scale cross-diffusion term for mass (momentum, energy) balance. This is because the aforementioned cross-scale coupling introduces meso-scale source/sink terms in the individual conservation laws identified by $r_T, r_H, r_M$ and $r_C$, respectively. The conservation laws must, therefore, be extended to close the equations and allow a multitude of THMC processes to occur simultaneously, which introduces cross-diffusion fluxes.

Here we derive the cross-diffusion approach from a different angle by following the coarse-graining operation. Techniques for the quantification of the uncertainty reduction through the coarse-graining operation are the main subject of statistical mechanics (Sethna, 2006). The basic principles can be discussed based on Shannon's entropy, Heisenberg uncertainty relationship, renormalisation theory, and the fluctuation theorem (Evans and Searle, 2002). For a more in-depth discussion and a broader perspective on the different techniques we recommend textbooks providing distillations of 50 years of statistical mechanics (Sethna, 2006) and material science applications (Balluffi et al., 2005). Here we present a functional analysis approach of probability theory applied to order parameter fluctuations. The first part of the analysis is often used in signal processing and image analysis (Buades et al., 2005). The following discussion, therefore, explains the convolution filter interpretation of the cross-diffusion waves discussed in part 1 (Regenauer-Lieb et al., 2020).

### 2.4.1 Gaussian Wavelet convolution as a blurring filter

In terms of uncertainty quantification the diffusion wavelet in Eq. 2 and shown in Fig. 7 is identified as a univariate Gaussian probability density function, also called a Gaussian kernel in image processing (Buades et al., 2005). Due to the linear nature of the partial differential equation we note that the following analysis is valid for any number of interactions. Fig. 7 shows an example of interaction of two diffusion wavelets with the probability distribution functions having an opposite sign of velocities



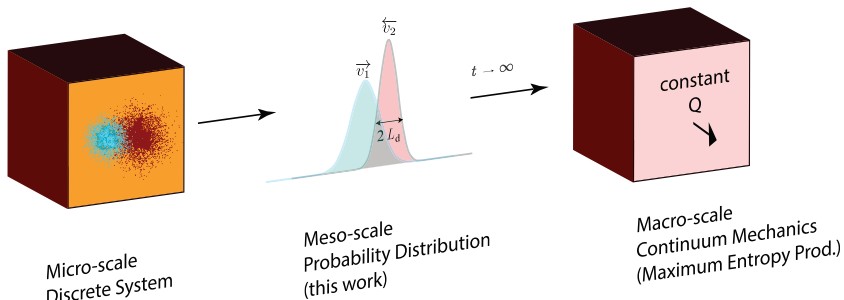

**Figure 7.** Consider a thermodynamic system as a planar surface from a macroscopic view. At the mesoscopic view, it is made up of subvolumes or more accurately subplanes in the current mathematical formulation. A mesoscopic subvolume is highlighted where two propagating diffusion wavelets interact. The discrete system represents the microscopic view where red and blue dots represent individual discrete chemical molecules or predators and prey or indeed generalised thermodynamic 'micro-engines' in an activated oscillatory quasi-steady state. We are interested in the assessment of the effect of the mesoscopic collision of micro-engines on the macroscopic system. From an equilibrium statistical mechanics perspective, we are therefore not interested in the local dynamics of molecules/micro-engines but use a convolution operation to describe the mesoscopic effect. This approach allows investigation of the meso-scale dynamics which feeds into the macro-scale continuum mechanics approach where sufficient time has elapsed so that the flux $Q$ becomes constant and the system has reached its maximum entropy production state (Collins and Houlsby, 1997). The system can then be described by the theory of thermomechanics, ie. the theory of thermodynamics with internal variables (Maugin and Muschik, 1999; Jacquey and Regenauer-Lieb, 2020). This approach also yields the familiar dissipative material properties/transport coefficients of the classical continuum approach. For constant thermodynamic force flux pairs we obtain for instance: the thermal conductivity of the material, for constant temperature/heat flux (T), permeability for a constant pressure/fluid flow pair (H), viscosity for a constant velocity/momentum flux pair (M), and the chemical diffusivity for a constant chemical potential/diffusion flow (C) pair (Oettinger, 2005).

implying a collision of the two wavelets due to their opposing direction of travel (Eq. 13).

$$f(x) = \frac{1}{L_{d(f)}\sqrt{\pi}} exp(-\frac{(x-\mu_f)^2}{L_{d(f)}^2})$$

$$g(x) = \frac{1}{L_{d(g)}\sqrt{\pi}} exp(-\frac{(x-\mu_g)^2}{2L_{d(g)}^2}),$$

(17)

where $\mu_f$ and $\mu_g$ are the function centroids of the the function $f(x)$ and $g(x)$ known as the mean in probability theory. Since

both wavelets are moving towards each other we would like to use a Lagrangian reference frame, here arbitrarily chosen to be $f(x-\tau)$, to assess how the shape of this function is modified by the passage of the function $g(\tau)$. The convolution operation is commutative, and we can also use the opposite reference frame. The convolution operation is defined by:

$$\int_{-\infty}^{\infty} f(x-\tau)g(\tau)d\tau = f \star g,$$

(18)





where $\tau$ denotes a translation in the positive $x$-direction. Due to the translational symmetry of the diffusion wavelet, it is

convenient to use the Fourier transform $F$ as the eigenfunctions of the translation $\tau$ operation are complex plane waves.

$$F(f(x)) = \exp(-\pi i k \mu_f) \exp\left(-\pi^2 L_{d(f)}^2 k^2\right), \tag{19}$$

where $k$ is the wave-vector of the Fourier transform. The wave-number and wave-vector are defined as $k = |\mathbf{k}| = 2\pi/\lambda$. The

convolution operation of function $f(x)$ and $g(x)$ in Fourier space is the product of the Fourier transforms $F(f(x))F(g(x))$.

Applying the same Fourier transform as in Eq. 19 to $F(g(x))$ and performing the product with $F(f(x))$ we obtain:

$$F(f(x))F(g(x)) = \exp(-\pi i k (\mu_f + \mu_g)) \exp\left(-\pi^2 (L_{d(f)} + L_{d(g)})^2 k^2\right). \tag{20}$$

The first exponential term is the mean amplitude of each Fourier wave-vector which can be decomposed as a real sinus-wave

and imaginary cosinus-wave that contains the phase information. The second term is a Gaussian probability density function

of the wave-vector.

This result is very useful as it can be used to analyse the behaviour upon the collision of two diffusion wavelets. The first

conclusion is that the convolution results in another Gaussian wavelet as can be seen from the comparison of Eqns. 20 and 19.

The second conclusion is that the convolved amplitude is reduced by the sum of the function centroids $\mu_f + \mu_g$ in the positive

$x$-direction and the breadth of the wavelet is enlarged by the variance $(L_{d(f)} + L_{d(g)})^2$ of the probability distribution function.

The result that the convolution operation reproduces a similar function to the original function is called self-similarity. The

Gaussian function is self-similar and defines a linear Gaussian scale-space where space-time relationships can be normalised

by the diffusion length. This is an important aspect of coupling across the scale. Another aspect of the Gaussian wavelet is

that the 1-D approach presented here can be extended and superposed to any dimension as the function is separable, i.e. it

can be applied to any dimension independently. In principle the technique can be extended to anisotropic diffusion, where

the introduction of a non-linear space varying Gaussian filter may lead to additional problems discussed next (Kichenassamy,

1997).

### 2.4.2 Gaussian Wavelet convolution as a sharpening filter

So far we have discussed the situation where the meso-scale convolution of the two diffusion wavelets appears as a smoothing

filter of colliding Gaussian wavelets, where high frequencies are quickly reduced and the high-frequency wavelet gets prefer-

entially diffused. This class of Gaussian low pass filter problems is not of interest for the nucleation of earthquakes as the small

scale local instabilities triggered by chemical reactions are filtered out. The opposite situation of a sharpening filter is, however,

very interesting. A sharpening filter is an expression used in image processing when applying a deconvolution operation using

a Dirac delta convolution kernel minus a Gaussian blur kernel. This is effectively an anti-diffusion filter, also known as an

uphill diffusion operation, which may lead to problems as it may violate thermodynamics without applying further restrictions

(Ulmer, 2010). This problem is well known in image processing for modelling anisotropic diffusion where it is known as the

Perona-Malik paradox (Kichenassamy, 1997).

However, the sharpening filter problem can be made thermodynamically consistent. The method has been described in part

1 (Regenauer-Lieb et al., 2020) through the consideration of meso-scale processes and the physics of acceleration waves that





radiate energy into the environment and regularise the local inconsistencies through their interaction with a stabilising constant large-scale thermodynamic flux. The communication mechanism between the meso-scale and macro-scale is a travelling acceleration wave. We have shown through mass and momentum balance that a local reaction term requests a local cross-diffusion

term. The approach in part 1 (Regenauer-Lieb et al., 2020) was based on a physics and mechanics of solids approach which essentially looks at the meso-scale negative diffusion process from a macroscale continuum mechanics perspective characterised by a positive diffusivity. Can we come to the same conclusion by looking at the meso-scale from a chemical-scale perspective by considering the mobility of atoms underpinning the meso-scale process in the light of the global changes of the free energy?

The observation of spontaneous unmixing from a fully diffused, intermingled, and unstable thermodynamic phase of two

different phases, e.g. solid and fluid (spinodal decomposition), is mathematically equivalent to an uphill diffusion problem. When interpreting the phenomenon through a linearised Fickian diffusion Ansatz (Eq. 2) we come to the perplexing conclusion that Eq. 2 has a thermodynamically inconsistent negative diffusivity (uphill diffusion). Note that the driving force for spinodal decomposition is not a chemical reaction but a reduction in the bulk free energy of the homogeneous mixture, which is thermodynamically unstable. The breakthrough in understanding this process in terms of a thermodynamically consistent

approach to the apparent uphill diffusion phenomenon of spinodal decomposition is attributed to Cahn (1961) and coworkers.

A detailed description of the approach is given in Balluffi et al. (2005). The solution is to consider a meso-scale length scale at the spinodal reaction front (hereafter called the spinodal) and express the diffusion process as a product of the diffusive mobility of atoms $M$, which must be a positive term to satisfy the second law and a gradient term of the large-scale generalised chemical potential. The gradient in the generalised chemical potential, in turn, is made up of a product of a meso-scale negative

second-order derivative of the free energy $f$ of the mixture concerning the concentration $C$ and a positive gradient of the concentration. This situation leads to a non-linear interdiffusivity $\tilde{D} = M \frac{\partial^2 f}{\partial C^2}$ which is negative at the spinodal.

$$\frac{\partial C}{\partial t} = \tilde{D} \nabla^2 \frac{\partial C}{\partial x}.$$   (21)

There are two ways to approach this thermodynamic problem which can be solved by considering the meso-scale thermodynamic couplings. The first approach is similar to the one presented in part 1 (Regenauer-Lieb et al., 2020) but developed

from a chemical perspective. For this, we decompose the nonlinear interdiffusion coefficient $\tilde{D}$ into $n$-coupled interdiffusion coefficients $\tilde{D}_{ij}$ expressed as a product of two matrices $L_{ik}$ and $\Upsilon_{kj}$ (Balluffi et al., 2005),

$$\widetilde{D}_{ij} = \sum_{k=1}^{N-1} L_{ik} \Upsilon_{kj}.$$   (22)

where $L_{ik}$ are the classical macro-scale chemical diffusivities, also known as the 'Onsager coefficients', and $\Upsilon_{kj}$ are the meso-scale thermodynamic factors that couple local chemical potentials to concentrations. For the specific case where the local,

meso-scale coupling coefficients are of mechanical origin, due to the effect of the volumetric strain triggered by a concentration change for instance, $\Upsilon_{kj}$ are known as the 'Vegard coefficients' (Balluffi et al., 2005).





For the least complicated multicomponent system the coupled interdiffusion matrix is

$$\widetilde{D}_{ij} = \begin{bmatrix} \widetilde{D}_{11} & \widetilde{D}_{12} \\ \widetilde{D}_{21} & \widetilde{D}_{22} \end{bmatrix}. \tag{23}$$

We will use the compact notation for the extended general multicomponent case and introduce the stability criterion for the interdiffusion matrix later. For the spinodal decomposition case an instability is triggered when the meso-scale couplings have a negative interdiffusivity $\tilde{D}_{ij}$ and a local spinodal wavefront develops. A negative interdiffusivity leads to an exponential growth of any perturbation exactly like in the previously discussed reaction-diffusion case.

For discussing the criterion for instability we follow the solution strategy originally suggested by Cahn (1961). For this we stay in the Fourier space and consider the general linearised diffusion problem:

$$C_i(x) - C_m = \int A(k) \exp(ikx)\, dk, \tag{24}$$

whereby $C_m$ is the average composition used to linearise $\tilde{D}$ into the components of the two phases. $A(k)$ are the amplitudes of each Fourier mode $k$ being

$$A(k) = \frac{1}{2\pi} \int (C_i - C_m) \exp(-ikx)\, dx. \tag{25}$$

Inserting the chosen interdiffusion couplings into the generalised solution Ansatz we can estimate the response of the system

through a perturbation analysis of the wave-vector in Fourier space whereby

$$\frac{\partial A(k,x)}{\partial t}) = R(k)A(k), \tag{26}$$

and $R(k)$ is the amplification factor. This generic perturbation analysis has the following solution:

$$A(k,t) = A(k,0) \exp(R(k)t). \tag{27}$$

This analysis divides the spinodal decomposition model into two areas. For the local scale around the spinodal composition

wave we expect positive values of $R(k)$ and the convolution of the two inter-diffusing phases is unstable and results in exponential growth of composition waves defined in Eq. 24, i.e. a sharpening filter with the maximum wave-vector $\frac{\partial R(k)}{\partial k} = 0$. Away from the spinodal composition waves, the amplification factor is smaller than zero ($R(k) < 0$), and the system is stable. Figure 8 shows three-time steps in the temporal evolution of a simple Cahn-Hilliard system illustrating dominant wavelength growth at the maximum amplification factor $R(k)$.

To visualize the spinodal wavelength changes in the system, we plot the normalized species concentration in the $x-$direction. The dynamics increases the spinodal wavelength as the system goes to the steady-state which drives the phase-separation process. For Fig. 8 we solve the Cahn-Hilliard equation using PetIGA in a 2D configuration with 64 finite elements in each direction. PetIGA is an open-source framework for high-performance computing that solves efficiently phase-field equations (Dalcin et al., 2016). The material parameters for the Cahn-Hilliard system allow for phase separation together with spinodal

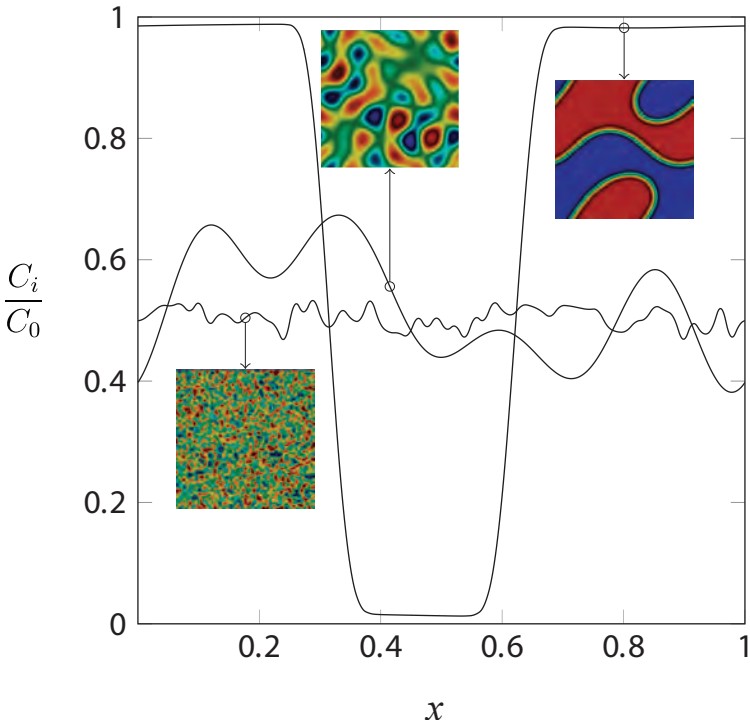

**Figure 8.** Cahn-Hilliard phase-field simulation of the spinodal decomposition controlled by the lowest wavenumber $k_{min}$ (longest wavelength) on the maximum amplification factor $R(k)$. The time and space-dependent species concentration profile $C_i/C_0$ is plotted as a function of the normalised domain $x$. The characteristic long wavelength pattern of the spinodal decomposition wave develops at time step three. Numerical simulations for the Cahn-Hilliard system are performed using a phase-field simulator described in Clavijo et al. (2019).

decomposition. Moreover, we use the isometrical analysis method to successfully discretize and solve the equation in its primal form. Such a method allows for high-order, highly-continuous basis functions (Cottrell et al., 2009).

For the generalisation to a multi-component system with $N$-coupled components, we evaluate the bi-directional response of each coupled system through a complex-valued wave function perturbation. This is also known as an order parameter perturbation analysis

$$i\frac{\partial \psi(k,t)}{\partial t} = -R(k)\tilde{D}_{ij}\nabla^2 A(k,t), \qquad (28)$$


where $i\psi(k,t)$ denotes the complex position-space wave function. Assuming $i\psi(k,t) = u(k,t) + iv(k,t)$, we recover the cross-diffusion type relationship of Eq. 24 in part 1 (Regenauer-Lieb et al., 2020) where the probability amplitude of the $u$-wave depends on the cross-diffusional coupling to the $v$-wave. Vice versa, the probability amplitude of the $v$-wave depends on the





cross-diffusion of $u$:

$$\frac{\partial u(k,t)}{\partial t} = -\tilde{D}_{ij}\nabla^2 v(k,t) \tag{29a}$$

$$\frac{\partial v(k,t)}{\partial t} = \tilde{D}_{ji}\nabla^2 u(k,t)\,. \tag{29b}$$

This bi-directional relationship between $u$- and $v$- wave, more generally known as the Kramers-Kronig relationship has provided a robust approach to investigate order parameters, phase transitions, and fluctuations (Balluffi et al., 2005; Sethna, 2006).

An important conclusion from this formulation is gained by a comparison of the partial differential Equations 29 and 15. The cross-diffusion problem leads at a quasi-steady state to a periodicity in space in the same way as the Lotka-Volterra oscillator model leads to a periodicity in time at quasi-steady state. The partial differential equations for cross-couplings are the same except that the cross-coupling coefficients in the Lotka-Volterra oscillator (Eq. 15) are reaction rates and the cross-coupling coefficients in the cross-diffusion formulation (Eq. 29) are diffusivities.

## 2.5 Cross-diffusion and its role in coupling of instabilities across scales

After discussion of the isolated spatial (cross-diffusion) and temporal (autocatalytic reaction) oscillators we can now extend the above approach for full space-time coupling for multiple reaction-diffusion equations and re-introduce reactions. An exemplary convolution of the mesoscopic oscillators is the self-similar Gaussian wavelet in space and time, which is also the Green's function of the local equilibrium statistical mechanics approach. In the linear case, any number of wavelets can be superposed, and a Fourier transform is an ideal procedure to capture the effect of multiple oscillators and local heterogeneities on the macro-system. In part 1 (Regenauer-Lieb et al., 2020) we proposed to generalise the cross-diffusion approach, which is well known in chemistry (Manning, 1970), to more general THMC terms defining cross-diffusion as the phenomenon where a gradient of one generalised thermodynamic force of species $C_j$ drives another generalised thermodynamic flux of species $C_i$, described by

$$\frac{\mathcal{D}C_i}{\mathcal{D}t} = \tilde{D}_{ii}\nabla^2 C_i + \sum_{j\neq i}\tilde{D}_{ij}\nabla^2 C_j + r_i \quad . \tag{30}$$

The species $j$ is identified as the cross-diffusion species other than the species $i$. Introducing a fully populated $(N \times N)$ diffusion matrix $\tilde{\mathbf{D}}_{ij}$, equation (30) can also be written as

$$\frac{\mathcal{D}C_i}{\mathcal{D}t} = \sum_{k=1}^{N}\tilde{D}_{ij}\nabla^2 C_k + r_i \quad , \tag{31}$$

whereby the classical (self-)diffusive length scale of each THMC process is defined by $\sqrt{4\tilde{D}_{ii}t}$. The cross-diffusion approach introduces also a new meso-scale coupling length scale that can provide a link between the large scale self-diffusion length scales of the THMC processes. In part 1 (Regenauer-Lieb et al., 2020) we have shown that the cross-diffusion length scale



$\sqrt{4\tilde{D}_{ij}t}$ is is defined by the kinetic material properties which can be evaluated by measuring the velocity of cross-diffusion waves in analogy to the generalisation of Eq. 11 for the Fisher-Kolmogorov wave. This leads to the equation for the wavespeed in Eq. 13 defined by the generalised Damköhler number and the cross-diffusion length scale $\sqrt{4\tilde{D}_{ij}t}$. Using the alternative
mechanical formulation the cross-diffusional wave speed turns out to be (Coleman and Gurtin, 1965)

$$v_{cross} = \sqrt{\frac{\mathbf{C}}{\rho}}. \tag{32}$$

where $\rho$ is the density and $\mathbf{C}$ is the $4^{th}$-order material stiffness tensor also known as the acoustic tensor discussed in part 1 (Regenauer-Lieb et al., 2020). The stability criterion of the diffusion problem is fully characterized by the diffusion matrix $\tilde{D}_{ij}$ with the diagonal elements $\tilde{D}_{ii}$ describing the normal self-diffusion and the off-diagonal elements the cross-diffusion processes
enabling coupling across scales. For consistency with the second law, all eigenvalues of the diffusion matrix must be real and positive, and hence the determinant of the diffusion matrix $\mathrm{Det}(\tilde{D}_{ij}) > 0$ as well as the trace of the diffusion matrix must be larger than zero. Complex eigenvalues of $\tilde{D}_{ij}$ result in oscillatory relaxation of any small perturbation to the equilibrium state, even in the absence of reaction (Vanag and Epstein, 2009).

Any local thermodynamic incompatibility that leads to complex eigenvalues of $\tilde{D}_{ij}$ must radiate energy in oscillatory in-
stabilities by so-called acceleration waves (Regenauer-Lieb et al., 2020), relaxing to the equilibrium state (Vanag and Epstein, 2009) to recover the second law of thermodynamics at large scale. This statement is at the heart of the nucleation of the cross-diffusional wave phenomenon. We will show in the following sections that the above described cross-diffusion formulation (Vanag and Epstein, 2009) can be extended to develop a generic multiphysics and multiscale THMC coupling approach to earthquake instabilities. This formulation implies a coupled cascade of THMC feedbacks over multiple diffusional length
scales honouring the reciprocal multiscale interplay of thermodynamic forces and fluxes.

By extending the diagonal diffusion matrix through the cross-diffusion coefficients in equation 31, a new cross-diffusion wave phenomenon is revealed that incorporates chemical waves at its lowest scale (Regenauer-Lieb et al., 2020). For the simple case of hydro-mechanical (HM) coupling, we have recently reported that the upscaled volumetric chemical strains can result in a hydromechanically coupled cross-diffusional pressure wave phenomenon (Hu et al., 2019).

The important element of cross-diffusion waves for earthquake physics is their capability to link one thermodynamic force with a thermodynamic flux at a different scale, thus synchronising the dynamics over vastly different diffusional time and length scales. This important aspect of earthquake physics was previously overlooked. The approach raises the possibility that dissipative waves can be detected before earthquake instabilities. Before discussing the potential earthquake application, we summarise observations from the laboratory and the field.

## 3 Laboratory and field observations of diffusion waves

So far we have discussed reaction-diffusion waves from an idealised continuum mechanics (Regenauer-Lieb et al., 2020) and a chemistry-based (this paper) viewpoint. We have addressed the uncertainty relationship which is implicit when going from a discrete to a continuum system. However, both viewpoints are based on mathematically ideal worlds. For real laboratory and





field applications, the processes are often occluded to direct observations, and the self- and cross-diffusion coefficients cannot

be easily derived. The verification or falsification of the reaction-diffusion "wave mechanics" approach is facing the following difficulties: (i.) Field observations offer a frozen-in snapshot of the dynamic process and may enable a direct identification of cross-diffusion length scales through observation of the microstructure. However, the assumption is that the reaction-diffusion wave has been given sufficient time to reach its maximum wavenumber as shown in Step 3 in Fig. 8. (ii.) Laboratory measurements offer insight into the dynamics, but they have the drawback that the microstructural processes at the meso-scale are

difficult to monitor directly (Schrank et al., 2020), and what is recorded is often an average response. This averaging requirement

some heuristic assumptions and simplifications. (iii.) The third and perhaps most important abstraction is that in the laboratory or field reference frame we most often see laboratory or earthquake instabilities as mechanical and not as a chemical, fluid, or thermal instability. The third problem can therefore only be addressed by exploring the solution space mathematically. This

paper proposes that earthquakes are a THMC convolution of all of these instabilities that cause volumetric and shear strains due to their different micromechanics. In the following, we will address the three problems sequentially using field and laboratory examples before exploring simple analytical solution applied to earthquake physics.

## 3.1 Diffusion waves at chemical scale

At first sight, we may expect that field examples offer the most direct access to the verification of the cross-diffusion wave

hypothesis, however, this is curtailed by following difficulties. The foregoing discussion on the Lotka-Volterra oscillator model compared to the spinodal composition wave highlights the fact that oscillatory responses can either stem from an autocatalytic reaction (Lotka-Volterra, oscillation in time) or an effective uphill diffusivity (oscillation in space) or both. In a geological field example we can investigate the frozen-in spatial response but cannot verify/falsify the theoretical expectations for the time evolution and hence, we cannot quantify the composite effect of the different wave speeds. From a mathematical point

of view we may expect two end-member cases. The first case is the temporal oscillator-case that is governing the system as in the Lotka-Volterra model (Eq. 15) and the other is the spatial oscillator-case that is governing the system in the form of waves fixed in space. This spatial oscillator is obtained by swapping the rate constants of the Lotka-Volterra cross-coupling rates by cross-diffusion coefficients as shown in Eq. 29. These two end-member cases are exactly what has been proposed to explain periodic patterns in the field (L'Heureux, 2013).

Accordingly, time-periodic chemical systems have been attributed to a mechanism discovered in reaction-diffusion experiments with hydrogels where colloidal precipitation patterns with a self-propagating rhythmic growth signature have been recorded (Liesegang, 1906). This special class of Turing precipitation patterns, encountered in reaction-diffusion systems, have been attributed to a large class of oscillatory pattern formation observed in chemistry, physics, biology (shells), medicine (gallstone, cysts, tumours, inflamed tissues) and geology (Nabika et al., 2020). In geology these oscillatory precipitation patterns

are encountered in a wide range of geological settings (igneous, sedimentary, hydrothermal and metamorphic) and classified as "Liesegang" patterns (L'Heureux, 2013). Rhythmic Liesegang patterns come in many guises and for widely different rock compositions. Figure 9 shows a typical set of Liesegang patterns encountered in a porous sandstone on the East Coast of Aus-



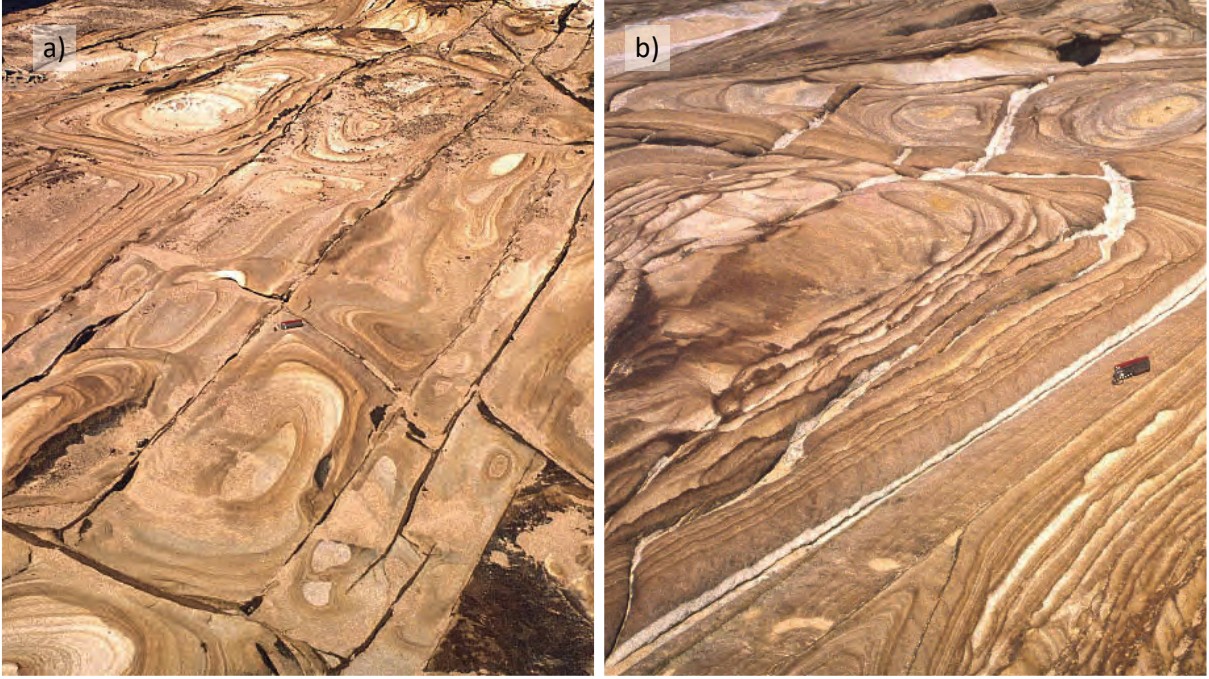

**Figure 9.** Liesegang patterns in Hawkesbury Sandstone, NSW where several sets of Liesegang patterns are observed. Siderite is a common constituent in the Hawkesbury Sandstone. After an oxidation reaction characteristic limonite, lepidocrite and ghoetite precipitation patterns are formed. These oscillatory reaction-diffusion patterns in the porous sandstone are often oblique to the sedimentary layering and commonly related to fractures which act as preferred fluid pathways. A possible explanation suggested by Ron Vernon (pers. comm.) is that oxidation reactions of siderite ($FeCO_3$) with percolating oxidized water through joints ($2FeCO_3 + H_2O + O \rightarrow 2FeO(OH) + 2CO_2$ ) have built the circular patterns shown in a) while a reduction reaction of percolating acid solutions ($2FeO(OH) + 2H^+ \rightarrow 2Fe_2^+ + 2H_2O + O_2$) may explain the bleaching around the joint in b). Photographs from Killcare, courtesy of Ron Vernon.

tralia. Although the Liesegang patterns form a striking pattern found everywhere in the coastal exposure in the greater Sydney area little fundamental work has been done on the investigation of these exposures. Comparisons with similar rocks found widespread around the globe have been used to propose a microbially induced oxidation of siderite by iron-oxidizing bacteria (e.g. Gallionella spp.) with following geochemically induced self-organisation of the reduced iron as a possible source for the pattern (Kettler et al., 2015).

Although Liesegang patterns have been encountered in many guises and more than 100 years of research on the subject has supplied a wealth of observations, a recent review (Nabika et al., 2020) reveals that the basic mechanism is still controversially discussed. Accordingly, the basic mechanism is attributed to two competing models classified by Nabika et al. (2020) as pre-nucleation and post-nucleation models. The pre-nucleation model describes a time-transient propagating dissolution-precipitation wave (L'Heureux, 2013) where the precipitate and depleted zones are frozen in at the nucleation and growth step. In contrast the post-nucleation model follows as a quasi-static pattern which emerges as a long time scale pattern after the





reaction-diffusion front has propagated through the medium. The latter model has been used to explain the origin and mech-
anism for banded iron formations (Wang et al., 2009) and striped zoning in agate (Wang and Merino, 1995). A more recent
study of mineralising systems interpreted as long-time scale solution of the Gray-Scott reaction-diffusion system is found in
Oberst et al. (2018). The Gray-Scott model neglects the meso-scale cross-diffusion term but considers self-replicating patterns
of the mineral system reproduced by inclusion of a non-linear source term. This approach therefore jumps straight to the long-
time scale solution as for the equivalent mechanical case of the Korteweg deVries - or the non-linear Schrödinger equations
discussed further below.

A universal mechanism for Liesegang patterns probably encompasses both end members in a mixture (Nabika et al., 2020).
This conclusion is supported by similarity of the Liesegang patterns to the spinodal decomposition models. In these models the
mathematical analysis of the Cahn-Hilliard spinodal composition wave reveals a surprisingly rich field of linear and non-linear
wavefronts with possible cross-overs from temporal to spatial wavenumber selection and wave speeds (Scheel, 2017). Such
a complex behaviour may be expected from the above discussion that includes the partial differential equation of a Lotka-
Volterra oscillator in time combined with the cross-diffusion oscillator in space. Due to the complexity of the behaviour we
therefore recommend to use the perturbation theory in terms of a Kramers-Kronig formalism (Eq. 28) for a full exploration of
the solution space. For direct analysis of sparse data provided by field observations such a complete analysis is, however, not
appropriate and simplifications need to be introduced.

In order to do so we need to perform laboratory experiments with analogue materials to gain insight into the timescales. For
such experiments, we may switch from a chemical perspective with fluxes of species to a mechanical viewpoint represented
by strain and strain-rates. Both can be coupled through an equation of state approach and mixture theory as discussed in part
1 (Regenauer-Lieb et al., 2020). Another approach is to jump from the meso-scale simulation directly to the continuum scale
through the introduction of an internal variable and the assumption of local equilibrium. This approach is also known as or
thermodynamics of internal variables (Maugin and Muschik, 1999; Jacquey and Regenauer-Lieb, 2020). We will discuss both
approaches in the next section.

### 3.2 Diffusion waves at mechanical scale

Oscillatory deformation bands are well documented in deformation experiments of plastic materials such as steel. Perhaps the
most well-studied effect is the development of characteristic Portevin-Le Chatelier (PLC) bands which show rhythmic bands
of volumetric strain in metals deformed under tensile conditions. The banding is attributed to a damped runaway effect induced
by a critical (negative) strain-rate sensitivity (Zaiser and Hähner, 1997). In PLC the transition from smooth to oscillatory
deformation is understood as a critical point phenomenon where mesoscopic fluctuations manifest themselves at the macro-
scale due to strain-rate softening.

A comprehensive review of theoretical approaches to model the phenomenon of an oscillatory material response of plastic
materials with a serrated stress-strain response (or jerky flow) can be found in Zaiser and Hähner (1997). According to the
review models first have been developed based on a phenomenological approach proposing that the effective stress applied at
the boundary of the deformed sample is a non-linear function of the strain rate $\dot{\varepsilon}$ which can become negative. In the following,





we will interpret the approach at the mechanical scale in the light of the reaction-diffusion approach and THMC coupling concerning the earthquake application.

### 585  3.2.1  Oscillatory deformation bands in plastic deformation of materials with internal structure

The phenomenological approach where the effective stress weakens with increasing strain-rate is ill-posed as it leads to a runaway effect when a small local perturbation in strain-rate grows without bounds in an ever-increasing manner. Various approaches have been proposed to address this problem. In the context of relating the phenomenological approach to a physics-based model, the most interesting approach is the one by Estrin and Kubin (1995). The authors proposed to regularise the ill-590  posed localised runaway strain by introducing a pseudo-diffusivity; the function that was proposed as an extension of Orowan's equation is

$$\dot{\varepsilon} = b\rho_m v_d + D\nabla^2\varepsilon, \tag{33}$$

where $b$ is the Burgers vector, $\rho_m$ the dislocation density, and $v_d$ the dislocation velocity and $D$ a diffusivity of the strain $\varepsilon$. This heuristic approach was later shown to be resolved as the 1-D solution obtained from the coarse-graining of the time-evolution 595  gradient flow dynamics of dislocations which results in a fractional reaction-diffusion equation (Monneau and Patrizi, 2012). The well-studied jerky flow phenomenon associated with PLC band formation is now understood as the result of initially independent, statistically distributed meso-scale dislocation mills ultimately coalescing and propagating into a macroscopic band. The best analysed mechanism for these mesoscopic fluctuations is dynamic strain ageing which gives rise to an additional characteristic retardation time scale to dislocation glide. Dynamic strain ageing operates by diffusing atoms (solute clouds) 600  that pin dislocations and temporarily arrest the glide dislocation segments. Therefore, the plastic strain rate cannot respond instantaneously to changes of the stress, and a meso-scale diffusion length/time scale emerges which stabilises the slip. The key to the oscillatory behaviour is the opposite strain-rate softening effect where the effect of retardation by diffusion is overcome through the increased mobilisation of additional dislocations (increasing the reaction rate) aided by thermal activation, hence increasing the disorder and thereby the average flow stress (Zaiser and Hähner, 1997). A similar effect of competing reaction-605  diffusion processes has been found in many other materials such as metal alloys (Brechet and Estrin, 1996) and self-oscillating polymer materials (Masuda et al., 2016). In an elegant discourse about possible earthquake nucleation mechanisms, Orowan (1960) postulated that earthquakes may indeed be triggered by the equivalent effect of an oscillatory response of creeping rocks at depth in the crust or mantle. Orowan (1960) drew the analogy of creep failure of an annealed steel where the stress oscillates between upper and lower yield stress forming oscillatory bands called "Lüders" bands. Plastic materials, in general, are capable 610  of arresting such small-scale fluctuations when slip tends to localise the meso-scale flow localisation and strain hardening ensures that the instability mechanism cannot go catastrophic. Orowan (1960) appealed therefore to the synchronising effect of thermal feedback at depths in the Earth's mantle where self-acceleration of creep is conceivable through an avalanche-like increase of deformation where shear heating occurs faster than conduction of heat from the shear plane, finally resulting in a localised melting event as an earthquake instability.

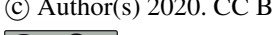



While runaway melting instabilities have indeed been postulated as a possible source mechanism for extremely deep earthquakes such as the 1994 great Bolivian earthquake (Kanamori et al., 1998), the mechanism may be considered an end member. Kanamori and Brodsky (2001) proposed that there are a great number of other possible earthquakes micro-instabilities without invoking a melting instability, however, the critical aspect remains to identify the physics of connecting small and large scales. If we consider THMC feedback as a source mechanism for earthquakes, we conclude that the largest scale coupling effect is
again the temperature. This becomes obvious from inspecting Table 1 where the thermal diffusion process defines the largest diffusional length scale. Thermal coupling is so efficient that a material can be considered macroscopically homogeneous and at thermostatic equilibrium, while at the mesoscopic scale it still shows widespread thermal fluctuations. If we refer again to the PLC-effect as an analogy of a thermally activated material it has been recognised that these statistically uncorrelated fluctuations may, for certain critical conditions, be coordinated in the shape of a thermal wavefront that propagates through the
material (Zaiser and Hähner, 1997). These wavefronts are typical acceleration waves illustrated in Fig 4 of part 1 (Regenauer-Lieb et al., 2020). While metals and rocks at depth are thermally activated materials, we need to discuss acceleration waves in brittle materials where the thermal activation process is less obvious.

### 3.2.2   Compression of rocks and rock analogues in the laboratory

Crushable granular materials show a similar strain-rate weakening behaviour as metals. This effect may lead to the phenomenon
of propagating compaction waves in crushed snow (Barraclough et al., 2017) and puffed rice (Guillard et al., 2015) where excellent experiments are available. The generic experimental configuration is shown in Fig. 10. Travelling compaction waves that reflect from the boundaries have been recorded by optical means (Guillard et al., 2015; Barraclough et al., 2017). Experiments have also been performed by partially soaking the puffed rice at the bottom of the experimental setup (Einav and Guillard, 2018). The interference of the compaction in unsaturated compaction and capillary-driven crushing of the puffed rice
led to oscillatory catastrophic events of global compaction with acoustic-emissions perceptive as loud audible beats termed "rice-quakes" by the authors (Einav and Guillard, 2018).

    An important aspect of the interpretation of the laboratory experiments is that, when going from a perspective of propagating chemical or thermal acceleration wavefronts to a recording of the mechanical response of the medium, a coarse-graining step is made. We are, therefore, in most cases, not able to record the expected individual multiscale THMC wavefronts but are more
likely to record the cumulative, convolved effect of the waveforms that underpin and support the mechanical deformation that can ultimately lead to macroscopic failure. From a mechanical perspective, we encounter two different dynamical regimes: in the first regime the dynamics of chemical, hydraulic, elastodynamic mechanical, and thermal wavefronts is still important, and the system cannot be simplified by a thermostatic assumption. The other regime is where the thermostatic assumption holds, and the problem can be described by a quasi-static framework, where the time dependence of the system can be neglected and
the problem reduces to an ideal plastic time-independent one, only controlled by the kinematic boundary conditions, e.g. the position of the porous ceramic platen applying the compression in Fig. 10.

    The thermomechanical theory of internal variables (Maugin and Muschik, 1999) is a variance of classical plasticity theory as it allows for additional time-dependent processes describing the evolution of local thermostatic equilibrium states through




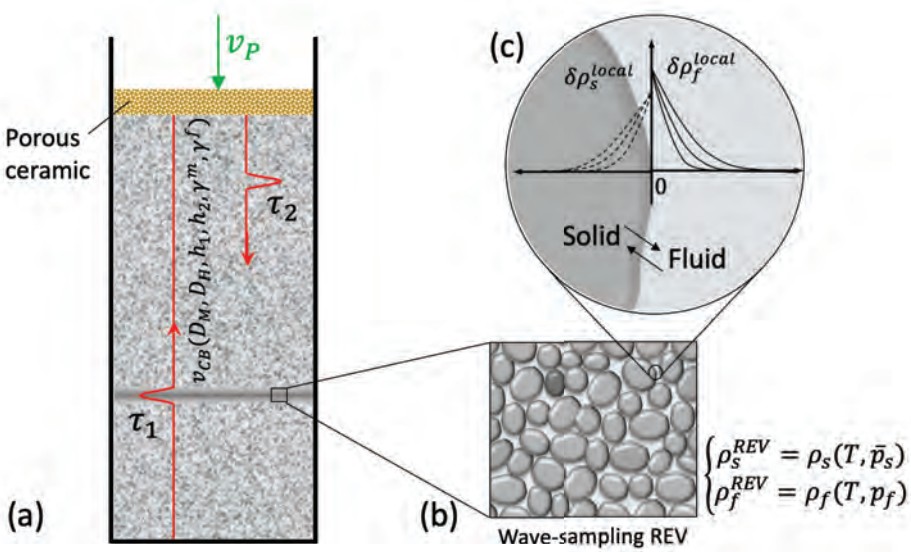

**Figure 10.** Experimental setup for crushing a granular strain-rate weakening medium. The problem is here formulated in analogy to the Terzaghi consolidation problem where a pore-filling fluid is diffusing out of the compacting region and crushed grains (solids) are replacing the collapsed pores. The meso-scale process is sampled by a wave-scale Representative Elementary Volume (REV).The macro-scale continuum-scale REV for a thermostatic formulation is defined in Fig. 2. The dependence of the registered wave speed on the applied boundary velocity is here suggested to be a possible diagnostic tool to differentiate between the thermostatic end-member and the full thermodynamic meso-scale formulation of the problem.

internal state variables. The introduction of additional time-dependent processes leads to a relative difference from the ideal

plastic kinematic reference frame, and the localisation band can move with respect to the static, ideal plastic solution which is fixed concerning the kinematic boundary conditions in a self-similar way. We can identify the two theoretical end-member cases by recording the dependence of the velocity of the optically recorded compaction waves on the velocity applied to the boundary of the experiments. If there is a positive correlation with the velocity applied to the boundary, the thermostatic end-member applies. Whereas if the velocity of the wave proves to be a material constant and independent of the boundary velocity,

the dynamic solution applies. This property can be used as a diagnostic tool for determining dissipative material properties as described in the appendix.

## 4   Application to earthquake physics

Theoretical approaches to the physics of earthquake instabilities originally were conceived as a shear instability on a frictional sliding surface (Brace and Byerlee, 1966). The role of pressure on the dynamics of the slider was derived empirically by

laboratory experiments defining a rate-and-state variable friction law (Dieterich, 1972, 1987; Ruina, 1983; Tse and Rice, 1986; Rice et al., 2001). Instabilities through shear heating feedback were later considered to play an important role (Ogawa, 1987;





Regenauer-Lieb and Yuen, 1998; Braeck and Podladchikov, 2007). Additionally, a thermally induced fluid pressurization term was found to be an important component for accelerated creep (Vardoulakis, 2001; Rice, 2006). Another important ingredient of the earthquake instability was thought to be the coupling of scales, where at least two different processes, operating at different time and length scales, interact (Ohnaka, 2003). The approach presented here summarizes these effects in the diffusion matrix $\tilde{\mathbf{D}}_{ij}$ (Eq. 31) and enables upscaling via renormalization group theory (Regenauer-Lieb et al., 2013b; Hanasoge et al., 2017).

### 4.1 Cross-diffusion waves

A fully populated diffusion matrix provides the opportunity to extend the postulate of coupling different scale processes in earthquake mechanics (Ohnaka, 2003). Cross-diffusion, in excitable media (Molotkov and Vakulenko, 1993), can lead to the formation of self-supporting standing wave solutions (Berenstein and Beta, 2012). The integrable focusing non-linear Schrödinger equation bears similarities but cross-diffusion waves describe a much broader class of waves and hence offer a richer solution for real-life applications (Tsyganov and Biktashev, 2014). It is, however, difficult to derive exact analytical propagating wave solutions of the cross-diffusion wave phenomenon. The piecewise linearised FitzHugh-Nagumo oscillator (Zemskov et al., 2017) is by far the best analysed prototype of a cross-diffusion wave triggered by an excitable system with short lived-spikes and a slow recovery. The FitzHugh-Nagumo cross-diffusion model has originally been developed to describe the propagation of electrical impulses in nerves (Antonioletti et al., 2017).

We use a generic application of this oscillator to illustrate the potential feedback of just two THMC processes and the conditions that may prepare the formation of a future fault plane. The real waveforms of the transient cross-diffusion waves are expected to be more complex, and further work is required for a full theoretical assessment of cross-diffusion waves (Tsyganov and Biktashev, 2014). We, therefore, present only a high-level application of wave solutions to transfer knowledge from other disciplines to this new field of research (e.g. mathematical biology, computational chemistry, ocean and ice waves, and photonics). We use analytical solutions developed in these fields, supplemented by our long-wavelength solution for the hydromechanically coupled case (Veveakis and Regenauer-Lieb, 2015). This case is currently being analysed in detailed laboratory experiments for testing the prediction of the cross-diffusion wave hypothesis. We, therefore, restrict ourselves here only to schematic illustrations and focus on an in-depth discussion of how the cross-diffusion concept may be applied to the earthquake phenomenon.

#### 4.1.1 Cross-diffusion waveforms

THMC cross-diffusion waves discussed in this work stem from multiscale fluctuations as possible wave sources with the superposition of multiple waves in a wide frequency spectrum. The convolution of these waves results in interesting dispersion patterns that bear similar characteristics to quasi-solitons encountered in optical systems where chromatic dispersion is strong leading to anomalous dispersion patterns that, unlike solitons, come in discrete portions (Paschotta, 2008). Quasi-solitons encountered in photonics have two oscillating tails, one going to the left with a different wavenumber to the one going to the right. If the amplitudes of the tails are small, quasi-solitons can be treated as slowly decaying real solitons which lose their energy by radiation to form the tail (Zakharov et al., 2004).



A classification of the solution of equation (31) for the more complex discussed case of cross-diffusion has been presented recently (Tsyganov and Biktashev, 2014), and a broader definition of the quasi-soliton wave has been adopted to cover the richer field of wave interactions encountered in nature. Quasi-solitons feature complex dispersion relationships, where the wave velocity of individual waves (phase velocity) have a different velocity to their smooth envelope wave groups (group velocity). This leads to dominant wave packet solutions akin to envelope solitons in the nonlinear Schrödinger equation shown

to be similar to the linear cross-diffusion equation in equation (31) (Tsyganov and Biktashev, 2014). They differ, however, from the classical soliton solution in that the shape and speed of such waves in the established regime do not depend on initial and boundary conditions and are fully determined by the material parameters of the medium that they travel through.

Wave-packet solutions emerge when the wave energy is concentrated around a finite wavenumber. When the dispersion is weak, the envelope travels with an approximately uniform group velocity. Linear cross-diffusion can create strong non-linear

dispersion, where - depending on the coefficients - three different wave types have been classified: (1) fixed-shape propagating waves, (2) envelope waves, (3a) multi-envelope waves, and (3b) intermediate regimes appearing as multi-envelope waves propagating as fixed shape most of the time but undergoing restructuring from time to time (Tsyganov and Biktashev, 2014).

The collisional behaviour of these classes of cross-diffusional waves is complicated. When the cross-diffusion wave hits an interface (including another cross-diffusion wave) the amplitude of the quasi-soliton changes, and there is a temporary

diminution of the amplitude or in extreme cases an annihilation. In most cases, they recover their original form gradually. This is another feature that is different from true solitons which do not change on impact. In two-dimensional systems, additional complexities arise as they can penetrate, break apart on collision, or reflect into different directions (Tsyganov and Biktashev, 2014). Zakharov et al. (2004) compare quasi-solitons with unstable particles in nuclear physics. This describes their behaviour on collision but does not capture their capability to coordinate into standing waves on long time scales (Regenauer-Lieb et al.,

2020). We propose that constructive collision of these waves may, in geological applications, lead to the bar code imprinted in the damage zone of seismogenic faults.

Of specific interest for the earthquake problem is the aspect of the fate of the accelerations carried by the waves (Regenauer-Lieb et al., 2020) for cases where the wave collides and collapse after the collision. In experiments carried out with collapsing puffed rice particles (Guillard et al., 2015) cross-diffusion waves were detected (Hu et al., 2019) which release an acoustic

emission when annihilating after collision with the top surface. We propose that for a homogeneous plastic zone as shown in Figure (**??**), the first wave-collisions on the symmetry axis of the future fault plane converts the energy loss of the cross-diffusion wave into local material damage that will act as a seed to future wave interactions and systematically grows the fault. For the case of heterogeneous materials, the collision of cross-diffusion waves with internal or bounding material surfaces would cause an alternative seed for the nucleation of earthquakes.

Another important aspect is the cross-scale coupling of the full spectrum of diffusion waves which may be seen as an energy cascade from extremely short wavelength chemical cross-diffusion waves to the longest wavelength which in most cases is modulated by the thermal and mechanical diffusion length scale as shown in Table 1. The spectral content of THMC wave interactions is extremely rich, and we have emphasized that it is perhaps best compared to a convolution sharpening filter for cases of instabilities where wave energy focusses on specific locations (Regenauer-Lieb et al., 2020).





The wavelength of chemical cross-diffusion waves is relatively short and because of their rapid decay, the cross-diffusion waves emanating from either side of the elastoplastic boundary may never collide in the centre. The largest wave energy that can be transferred by cross-diffusion waves into the future fault plane is expected when the entire THMC sequence cross-diffusion phenomenon is triggered over eight to nine orders of magnitude for serpentinite ETS instability which can be modelled by the Lotka-Volterra type model (Poulet et al., 2014b).

### 4.1.2   Cross-diffusion collisions


In the context of an earthquake instability, shorter wavelength cross-diffusional waves might cascade to a small dispersion limit of a coordinated long-wavelength instability around a dominant wavenumber with a maximum amplification factor $R(k)$ (see Eq. 26). Recently, a cascade mechanism for coupling seven orders of the length scale of quasi-solitary waves has been discovered in optical fibres (Eberhard et al., 2017a). This resonant-like scattering mode may arise through incident cross-

diffusional waves colliding with reflected cross-diffusional waves in a finite width around the future fault core. On the grounds of the above theoretical considerations, we postulate a series of evolutionary steps that may lead to a macro-scale rogue-wave earthquake instability. These are:

   – **Step 1** (illustrated in Figure 11); Upon ongoing geodynamic loading a zone may form, where the rock is loaded past the elastic limit, and the plastic yield stress of the rock is reached. This defines the system size for cross-diffusion waves.

– **Step 2** (illustrated in Figure ??); If the criterion $R(k) > 0$ is met, cross-diffusional waves are expected to nucleate on each elastic-plastic boundary and propagate to the centre plane. In the hydromechanically coupled problem, these waves are propagating fronts of internal volumetric fluid-solid mass exchange and appear as local fluid overpressure. For a perfectly homogeneous plastic zone the first collision of cross-diffusion waves would be expected in the centre plane.

   – **Step 3** (illustrated in Figure ??); Collision of cross-diffusion waves can sample energy from the environment into a

central 'rogue wave' event which for cross-diffusional $P$-waves discussed here are a sudden spike in pressure. The energy for this spike is sampled from the amplitude of the oscillatory tails of the cross-diffusion waves through an 'uphill-diffusion' into the future central fault plane. A resulting fluid pressure spike on the future central fault plane is here interpreted as a possible trigger for the earthquake event through a double-couple mechanism on the fluid-lubricated fault planes. If the shear displacements can be quantified, we may be able to use the long-timescale standing wave solution

of the cross-diffusional $P$-wave as a geological interpretation tool of the earthquake event as a characteristic 'bar-code' like feature in the damage zone around the fault-plane is expected to contain valuable information about the earthquake source. Figure ?? shows the long-time scale Korteweg-de Vries standing wave solution (Veveakis and Regenauer-Lieb, 2015) where the characteristic wave-tails of the cross-diffusion waves have vanished entirely. A possible geological application of the Korteweg-de Vries standing wave solution is discussed in Regenauer-Lieb et al. (2013a) where Fig.

23 shows the scaling of the spacing between the pressure spikes with the characteristic dissipative parameters.





**Step 1 (t = 0)**

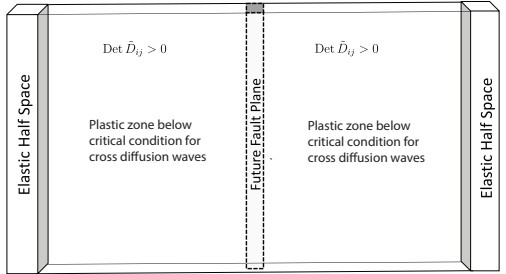

Establishing a plastic zone for the future fault damage zone

**Step 2 (critical condition for waves met)**

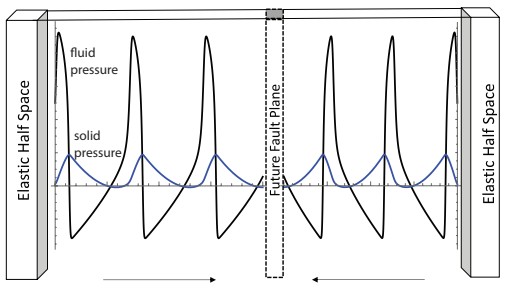

Direction of traveling cross-diffusion waves from both sides

**Step 3 (earthquake event, on fault plane)**

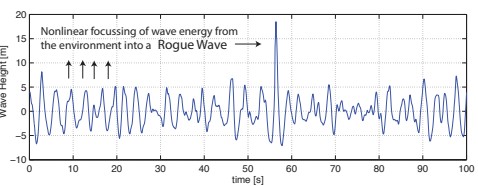

Wave height record of the Draupner Platform Rogue Wave event

**Step 4 (possible postseismic pattern)**

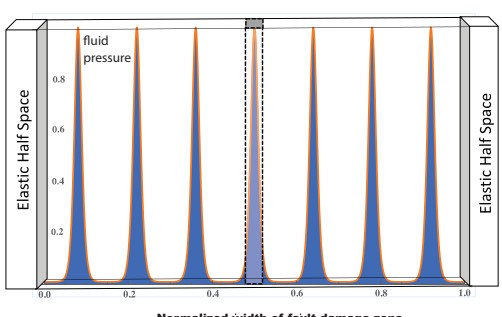

Normalized width of fault damage zone

**Figure 11.** Proposed earthquake cycle: Step 1) The size of the plastic zone is dependent on the boundary conditions and the micro-physics accommodating plastic deformation. We assume in our discussion that the zone is deforming pervasively by a power-law. Step 2) For critical conditions ($R(k) > 0$) excitable self-oscillatory cross-diffusion waves propagate from opposite sides towards the future fault plane. Here, we use the analytical solution of the piecewise linear approximation of the FitzHugh-Nagumo cross-diffusion wave (Zemskov et al., 2017) for illustration. A snapshot is shown just before the collision on the future fault plane. The pressure wave of the two phases $C_1$ (fluid, black) and $C_2$ (solid, blue) is shown in the graph. The collision of the waves can lead to an extreme sharpening of the waveform collecting wave-power over eight to nine orders of magnitude into a rogue wave event. Step 3) Rogue wave event on the Draupner Platform in the North Sea (Cavaleri et al., 2016). Step 4) The cross-diffusion waves can turn into real solitons if the amplitude of their radiative tails becomes vanishingly small. This situation may follow the rogue-wave event during relaxation or during prolonged slow deformation without rogue wave. Under these conditions the lowest wavenumber $k_{min}$ resonant-like scattering P-wave mode can fully develop, and the static solution can be reduced to lower dimension by considering only the self-diffusion coefficients. A very simple solution is the Korteweg-de Vries equation (Regenauer-Lieb et al., 2013a; Veveakis and Regenauer-Lieb, 2015) where a strictly periodic pattern of overpressured fluid channels (non-dimensional overpressure labelled on the ordinate) follows. The dynamics associated with earthquakes results, however, more likely into more complex patterns such as those described by the nonlinear Schrödinger equation. Slow deformation or slow slip events might be better candidates for the solution shown in Step 4. A good example is the periodic fracture pattern stemming from slow diagenetic reactions in unconventional shale gas reservoirs or coal cleats in coal seams (Alevizos et al., 2017) which do not need an earthquake instability for formation.





## 5    Discussion and conclusions

We have introduced a new approach to THMC instabilities using the concept of cross-diffusion. To simplify the application, we have so far only focussed on dissipative cross-diffusional $P$-waves. We may now discuss the inclusion of cross-diffusional $S$-waves and speculate about the interaction with the $P$-waves. We have shown (Regenauer-Lieb et al., 2020) that the wave equa-

tions can be decomposed by using the Helmholtz decomposition of the equation of motion, into $P-$ (compressional) and $S-$ (shear) cross-diffusion waves. The analysis suggests that cross-diffusion provides a crucial link to allow cross-scale coupling of the multiphysics processes potentially leading to earthquake instabilities. For critical cross-diffusion coefficients and associated reaction terms, a bi-directional energy cascade of acceleration cross-diffusion $P$-waves, from small scale thermo-chemical (TC) and chemo-mechanical (CM) dissipative waves to meso-scale hydro-mechanical (HM) dissipative waves, can be triggered

by a geodynamic driving force. This in turn may nucleate shear-cross-diffusion waves in the form of a thermo-mechanical (TM) shear-wave instability.

The multiscale THMC-waves initially are low-energy release volumetric diffusion $P$-waves shown in Step 2 (Fig. 11) and are free of kinetic energy, but if they trigger thermo-mechanical cross-diffusion shear waves (diffusion S-waves), they can tap into a significant portion of the stored elastic energy around the fault. This process may, for a critical set of reaction-diffusion

parameters, ultimately lead to a substantial energy release sufficient to communicate via cross-diffusion with their elastic counterparts (Cartwright et al., 1997), and an earthquake instability occurs. The rogue cross-diffusion $P$- wave may thus be understood as a trigger to lubricate the earthquake fault. The cross-diffusion $S$-wave may by itself develop a similar cascade of energy and create its rogue wave, which is the earthquake instability itself.

We have also presented in Step 3 (Fig. 11) the iconic Draupner Wave example recorded on 1 January 1995 by a downward

looking laser beam on the oil platform. The exceptional nature of these rogue waves bears many similarities with earthquake events in terms of predictability. A hindcast analysis of the Draupner wave came to the conclusion that rogue waves are not predictable with current statistical means (Cavaleri et al., 2016). Records at a specific location are often misleading and the probability of detecting a rogue wave must be considered both in space and time; they have thought to be accepted as part of the reality of the ocean at a given sea state (Cavaleri et al., 2016). Recent work refutes this conclusion and raises hopes that a

statistical analysis is possible based on large deviation theory combined with the simplified non-linear Schröedinger equation solution of the cross-diffusion waves using random initial data (Dematteis et al., 2019). This raises hopes for earthquake research, acknowledging that it took nearly 15 years of research on the Draupner Wave to come to this point. To make progress on this matter, we have to leave the analytical assessment of the problem presented in this paper aside and investigate numerical solutions.

Numerical solutions of this stiff problem are unfortunately difficult, and it may be useful to consider weak-formulations by a reduction procedure through adiabatic elimination of the fast cross-diffusion process into an effective cross-diffusion formulation by time integration to the slow self-diffusion time scale (Biktashev and Tsyganov, 2016). Specifically, the upscaled solutions can be regarded as long-wavelength (infinite/quasistatic time-scale limits) of the travelling cross-diffusion wave solutions. The dominant wave number is described by a critical ratio of effective self-diffusion coefficients. The relationship





between the long-wavelength soliton cnoidal solution in Step 4 (Fig. 11) of the Korteweg-de Vries equation (Regenauer-Lieb et al., 2013a; Veveakis and Regenauer-Lieb, 2015) and the quasi-soliton solution with oscillating tails can be derived by the Jacobi elliptic functions method of solution. If the modulus of the Jacoby elliptic function asymptotically approaches zero the solutions shown in Step 4 (Fig. 11) can be obtained (Wang et al., 2018).

The quasistatic spatially inhomogeneous solutions of Step 4 (Fig. 11) are also known as Turing patterns (Turing, 1952). The
special role of cross diffusion for the emergence of Turing patterns is their capability to generate spatial periodicity (Berenstein and Beta, 2012), an aspect that is often overlooked in the analysis of Turing patterns. The upscaled Korteweg-de Vries equation does not necessarily honour the co-dependence between the diffusion and reaction rate constants of the cross-diffusion process, and results need to be investigated with caution (Hu et al., 2019; Vanag and Epstein, 2009). Turing patterns appear because around the bifurcation point in the reaction-self-diffusion system, linearly unstable eigenfunctions exist that grow exponentially
with time. To regularize the problem, finely tuned non-linear terms need to be introduced in the reaction-diffusion equation (such as in the nonlinear Schrödinger equation) or, more conveniently, linear cross-diffusion-like terms need to be added in a coupled system of equations. We postulate here that THMC-Turing patterns are indeed the multiscale patterns observed in nature (Sethna et al., 2001). They offer themselves as an ideal tool for inversion of the effective self-diffusion THMC coefficients and their implicit reaction rates. Interpreting geological structures in terms of these process parameters will allow
identification of principal processes underpinning the earthquake mechanism.

Further theoretical considerations on the nature of cross-diffusion waves may also help in the design and analysis of laboratory experiments. Quasi-solitons are similar to real solitons in that they can penetrate through each other and reflect from boundaries. Differences are that the amplitudes of the true solitons do not change after an impact while the dynamics of quasi-solitons on impact are often naturally seen as a temporary diminution of the amplitude with subsequent gradual recovery
(Tsyganov and Biktashev, 2014). Another important difference is that the amplitude and speed of a true soliton depend on initial conditions, while for the quasi-soliton they depend on the material parameters. This property offers new avenues for earthquake physics. We are currently investigating these wave phenomena in controlled laboratory experiments and attempt to use the unique relationship of amplitude and wave speed dependence on material properties as a diagnostic tool for data assimilation.

# Appendices

## A  Derivation of dissipative material properties from laboratory measurements

The velocity measurement in the laboratory requires an identification of a relative reference frame. When shifting from a material reference frame inside the rock formation, as discussed before, to a laboratory or field reference frame we have to consider the relative direction of the travelling wave and subtract or add the velocity vector of the boundaries of the experiment
to calculate the material velocity vector of the wave from a video recording. In practice, the material wave velocity is expected





to be very much faster than the background geological strain rate in the field or the velocity of the boundary, and the correction is only a minor one and may be neglected (Guillard et al., 2015). However, there exists a theoretical limit of vanishing wave speed of acceleration waves in which this is important (Barraclough et al., 2017). This limit underpins the classic theory of localisation phenomena in plasticity (Rudnicki and Rice, 1975), where the acceleration wave speed is set to zero (standing soliton wave)

and acceleration waves transform into stationary localisation bands. Although, as emphasized before, the comparison with conservative systems is inappropriate, we can compare this situation with a resonance phenomenon in elasticity. In the case of a diffusion wave, this situation is equivalent to the diffusion wave having the capability to retain the quickly fading information of the boundary conditions and convolve constructively with the reflected wave from the opposite boundary. This standing soliton wave solution is only possible for long wavelengths, low-frequency end members due to the physics of high-frequency

damping. The standing wave phenomenon is identified as the final case of a (thermostatic) travelling wave solution with zero velocity.

   If localisation phenomena in the form of travelling waves or standing waves can be observed, a new diagnostic test for deriving dissipative material properties from laboratory experiments can be suggested. This test reveals how close the experiment is to thermostatic equilibrium. If the wave is controlled by its own dynamics and no relationship between the applied background

strain rate of the experiment (e.g. velocity of the piston) applies, the wave velocity can be approximated by the square root relationship of the Fisher-Kolmogorov type (Eq. 13). The reason for the independence of dynamic waves from the boundary velocity is that the waves quickly loose the information of the initial conditions imposed by the step function applied at the boundaries of the experiment. These travelling wave solution are a new type of waves having an extremely rich dynamics. They are quasi-solitons with very different properties to the classical soliton waves (Tsyganov and Biktashev, 2014). On the other

hand, for the case of near thermostatic equilibrium, this relationship should change to a dependence on the applied boundary velocity, as the information of the applied boundary condition can spread through the sample and the diffusion wave has achieved a constructive convolution of the waves. An extreme quasi-static, thermostatic solution is shown in Step 4 (Fig. 11) where a stationary (standing) wave is shown with zero velocity. The derivation of dissipative material parameters for the zero velocity case has already been discussed (Regenauer-Lieb et al., 2013a). For the thermostatic case the wave velocity should be

zero or be dependent on the boundary velocity, as discussed next. This suggested diagnostic tool opens a new way to quantify dynamic material properties and to identify how close the system is to thermostatic equilibrium.

### A.1 Thermostatic soliton waves

Classical plasticity theory can be extended to account for the slow dynamics before approaching the full quasi-static case. For this case, we may neglect the dynamics of forward and backward reflecting waves from the boundaries and introduce

an internal state variable that captures the relaxation to quasi-static equilibrium (Jacquey and Regenauer-Lieb, 2020). The dynamics are assessed by assuming a meso-scale thermostatic equilibrium state that is linked to the next local equilibrium state by a finite-rate process slaved to the time scale of the irreversible processes (Maugin and Muschik, 1999). This is captured by





the characteristic dissipative time scale of the selected internal variable $\alpha$ which defines the relaxation to the equilibrium of the time-dependent processes. This time scale may be defined as

$$t_\alpha = \frac{\alpha}{\dot{\alpha}}, \tag{34}$$

and the velocity of the thermodynamic wave of internal variables may hence be identified as

$$v_{(thermostatic)} = \frac{L_d}{t_\alpha}. \tag{35}$$

For the thermostatic case the experiment records a wave velocity that is dependent on the applied boundary velocity and we can use the above discussed theory of internal variables. For this case material parameters can be derived through measuring
the diffusive length scale $L_d$ (e.g. the thickness of the propagating compaction band) and the background strain rate as $\dot{\alpha}$ (e.g. compaction strain-rate imposed on the sample by the compression piston). This interpretation assumes that all deformation activity is taking place in the active band. We can invert the magnitude of the internal variable $\alpha$ (e.g. the compaction strain) from measuring the material speed of the propagating compaction band by using the mass conservation criterion in Eq. 35. This method of deriving the band strain in a propagating compaction band in compaction of snow has been used by Barraclough
et al. (2017).

We have performed a similar compaction experiment with a highly porous limestone which produces first a stationary compaction band roughly in the middle of the sample after only 3% axial strain (Chen et al., 2020). When interpreting the result in terms of a standing wave limit (zero acceleration wave speed) we expected the location to be strictly linked to the quasi-static geometric constraints which can be either the size of the sample or an internal defect. Our time-lapse X-ray Microscopy
experiment (see Chen et al. (2020) for details) revealed a local porosity variation as the initial seed of the compaction band. In subsequent stages of the experiment, the stationary discrete compaction band in turn acted as the nucleation point for a compaction wave travelling downwards from the discrete band ultimately leading to full compaction of the lower part of the sample. This observation is strictly identical to the experiments performed by Barraclough et al. (2017).

### A.2 Dynamic quasi-soliton waves

In the case where the experiment records a wave velocity that is independent of the applied boundary velocity, we can derive the dissipative material properties directly from measuring the diffusion length scale $L_d$ and with the measured material velocity inverting Eq. 13 to obtain the effective Damköhler number $Da$. Alternatively, we can derive the instantaneous modulus $\mathbf{C}$ from inverting Eq. 32 from a measurement of the density $\rho$ and the band velocity $v$.

Guillard et al. (2015) have performed experiments with compacted puffed rice in a setup similar to Fig. 10 recording the
alternative case of a dynamic solution. The wave speed was found to be independent of the applied boundary velocity. Guillard et al. (2015) found that, when varying the velocity by a factor of 2.5, the number of propagating bands increased but not their velocity. If the density of the travelling compaction band can be measured by high-speed (Synchrotron) X-Ray-CT or radiography the instantaneous modulus $\mathbf{C}$ can directly be inverted from the band velocity. Such a measurement could then be used to derive the effective elastodynamic modulus quantifying the visco-elasto-plastic properties of the material.



**6  Acknowledgments**

This work was supported by the Australian Research Council (ARC DP170104550, DP170104557, LP170100233) and the strategic SPF01 fund of UNSW, Sydney. We would especially like to acknowledge the thought provoking experimental work of Francois Guillard and Itai Einav, as well as Barraclough et al. which may provide a new method for identification of dissipative material parameters. Further laboratory work is urgently needed to explore this new avenue.



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
