# Peer review of "Cross-Diffusion Waves as a trigger for multiscale, multiphysics Instabilities: Application to earthquakes"

_Solid Earth, 2020_

## Referee Comment (RC1) · Angelo De Santis (Referee) · 8 Mar 2021

The paper discusses the THMC reaction-diffusion equations as underlying equations for the instabilities preceding earthquakes. I understand the efforts and troubles the authors met in preparing this paper: the topic is new for seismology and the theory grown in other fields had to be fit in the seismological frame. I admit that the paper results very difficult to follow in some parts. From one side, some parts are didactical (e.g. 2.1, 2.3), others are technical (2.2), other simply colloquial (section 4): and this dichotomy is more evident because the latter concerns the main topic of the paper. Anyway, I found the paper very interesting and plenty of fresh indications for earthquake

physics, with the presence of such as cross-diffusion waves and stationary waves. Nevertheless, I have two main concerns. The first regards the fact that the general theory overcomes the application to earthquakes very much. In this aspect the paper should be more balanced, in view of the fact that is dedicated to the application to earthquakes. My second concern is that the stationary waves have never been clearly found and neither weakly detected by earthquake seismology (apart from presumed but disputed findings present in Russian literature). If this is true, the entire presented framework, although intriguing and compelling, will be relegated only as a curiosity within a theoretical corner without any real application. This could be the most negative aspect of this work. My suggestion is to ask the Authors to work on the paper in order to make a "moderate" (more than minor, less than major) revision in order to let some concepts and their relationships more clear. In the following, I list also some more punctual indications.

Lines 1-10 Abstract. I found it very short with respect the many information given in the main text. I suggest to expand it in order to include something more. By the way, the same indications (with the due adaptions) can be given for the section 5 of "Discussion and conclusions". See also below.

Line 28. I suggest Crampin et al. 2013 instead of Crampin and Gao 2015, as it takes the problem with a more seismological point of view, proposing the solutions for four typical seismological conondrums.

Lines 28-29 (also 64). I suggest the term "self-affinity" instead of "self-similarity".

Line 29. Better "log frequency- log magnitude relationship"

Line 34. "standing waves": how different are they from the "KaY waves" (Yagodin, 2017), whose existence is highly disputed (e.g. Koronovsky et al. 2019)?

Line 52. "holistic way...": how far is it from the "geosystemics approach" (De Santis 2009, 2014)? Probably they both share the same foundations.

Lines 106-108. It is said "... to first order ..." but the equation (2) is of the second order in space. Caption Figure 3, first line: please correct "sale" with "scale".

Table 1. It is not fully clear how the given values of the table have been estimated.

Line 237. You mention the "autowaves" but you provide only one reference (Antonioletti et al. 2017). Are there some more references?

Line 279. "... with some apparent regularity in their recurrence ...", in the sense that the regularity is only apparent: the 2004 Parkfield event skipped the previous apparent periodicity.

Line 280. Please remove one bracket after "1997"

Line 292. Better "quasi-periodicity in space"?

Line 511. The line is broken. In addition, the present sentence is not clear.

Lines 569-570. The sentence "This approach ..." is not clear

Lines 721, 745, 749, 757. There are "???" for some figures.

Line 760. Pag. 31. Where is "Step 4" in the main text that is mentioned in the Figure 11 caption?

Lines 779-789. I think this part should be better exposed, because concerns the predictability or not of the cross-diffusion (here called rogue) waves. The conclusions are not clear: what I understand is that we should "leave the (present) analytical assessment ... aside" and pass to "investigate numerical solutions". But the next section (lines 790-798) poses many limits to this step.

Lines 895 and next ones. Web links are mostly double. Please check them.

Line 974. Why is the title of this reference in capital letters?

References

De Santis A., Geosystemics, Proceedings 3rd IASME/WSEAS International Conference on Geology and Seismology (GES'09), Cambridge, 36-40, 2009.

De Santis A., Geosystemics, entropy and criticality of earthquakes: a vision of our planet and a key of access, in "Nonlinear phenomena in Complex Systems: from Nano to Macro Scale" ed. E. Stanley and D. Matrasulov, NATO Science for Peace and Security Series – C: Environmental Security, 2014.

Koronovsky, N.V., Zakharov, V.S. & Naimark, A.A. Short-Term Earthquake Prediction: Reality, Research Promise, or a Phantom Project?. Moscow Univ. Geol. Bull. 74, 333–341 (2019). https://doi.org/10.3103/S0145875219040057

Yagodin, A.P., International Center of the Earthquake Prediction, 2017. https://sites.google.com/site/earthquakepredict/r1

---

## Referee Comment (RC2) · Anonymous Referee #2 · 10 May 2021

I had high expectations for this manuscript, having reviewed the "Part 1" paper of this two-part series. The premise for this current manuscript is that rogue fluid pressure waves can be quantified via cross coupling of off-diagonal diffusivity terms in reaction diffusion partial differential equations, and such waves are earthquake triggering. This is a companion paper to "Part 1" which described the general instability and coupling features. The paper offers a great high-end discussion of earthquake physics from the perspective of THMC processes across time and length scales and as such, it seems at the start of the paper that it offers a good overview and thus a good inclusion in the special issue. Placing an important instability such as earthquake triggering in the

context of broader THMC instabilities is certainly an interesting precept for a paper.

However I think the paper frequently veers off course, and during a first read every time I was met with a promise of discussing earthquake physics the paper lumbers off onto barely related topics (Lotka-Volterra chemical oscillators, solitons, dislocation theory, spinodal decomposition, and on and on.) Perhaps the paper would be more readable if these sections were omitted or at least shortened (we don't, for example, need a lengthy discussion on wave propagation, predator-prey problems, and spinodal decomposition just to introduce cross terms in a diffusion matrix– this makes me think that part of this paper were in fact prepared originally as a textbook). Having already reviewed "Part 1", I was really looking forward to the meat of this paper which is the mathematical workup to the cross diffusional rogue wave treatment in the context of earthquake instability, where much of this ancillary discussion would have already occurred in Part 1. It is perhaps too late to consider this option (i.e. placing the background of THMC instabilities in paper 1, leaving paper 2 to focus on earthquakes). But that would be the ideal situation.

The linking of chemical oscillators to rogue waves as presented however is unfortunately ad hoc at best: although a summary of reaction-diffusion instabilities and related phenomena like Leisegang bands is perhaps an appropriate summary for a journal issue devoted to coupled THMC modeling, frankly the Leisegang phenomena and its related dynamics have little to do with cross diffusional terms and in particular with rogue waves. It seems the paper relies on the facts that both are "waves" to make a similarity argument, despite the inherent differences in the physics. The discussion on deformation bands is certainly appropriate but again the discussion loses robustness when linking to dislocation processes. It would do the paper well to focus on earthquakes and cross diffusional phenomena as the title suggests and forgo all this extra, unnecessary, and frankly unrelated stuff.

I am no expert in the propagation of electro-cardio waves and the associated instability, but the arguments presented in this paper would be better served if the so-called cross

diffusional approach used to describe this (FitzHugh-Nagumo oscillator) were more fleshed out (section 4.1), rather than mentioned in passing. Time spent on analogy between the FitzHugh-Nagumo mechano-electro-chemical oscillator would do much to improve the paper and not the Volterra chemical wave dynamics which owe nothing to cross diffusional terms. By the time the paper starts actually discussing the earthquake instability, there are only a few pages left with only a disappointing arm-waving level of treatment. I did not read the Discussion and Conclusion section.

I think a paper linking seismically observed "tremors" as fluid-motion, rogue wave physics, the interesting rice-krispie earthquake-like experiments, discussions of compaction waves, direct application of coarse-graining, and perhaps the FitzHugh-Nagumo oscillator (if at all applicable) would be a readable paper appropriate for this special issue. The rest just seems like unnecessary fluff. The appendices are unnecessary. I would recommend a rewrite along these lines.

Some additional comments:

1. Line 1/abstract - "Instabilities appears twice - change second occurrence to "source dominated source mechanisms". 2. Line 8 "These are here interpreted as a trigger. . ." 3. Line 13 "In this paper (Part 2) we investigate. . ... 4. Line 17 "Patterns in Our Planet" probably should be italicized 5. Line 23 – capitalize "Part 1" here and throughout the paper 6. Line 43 – don't capitalize "earth" 7. Line 63 – add coma after parenthesized term 8. Line 92 – does the Regenauer et al. 2020 paper refer to "Paper 1"? If so this reference needs to be updated. 9. Page 5, Figure Caption to Figure 2 – should provide a reference or two to the coarse graining discussed in the figure – (does this figure come from another reference? If so it should be treated accordingly. 10. Line 108 – should the linearized form of equation (1) contain a different notation for the "linearized" reaction rate expressed in equation 2? 11. 11. Line 110 – should reference Table 1 here. 12. Line 170 – eliminate the first comma 13. Line 183 Eliminate the comma 14. The paper would lose nothing in range and scope of content if Figures 3-8 were eliminated, along with accompanying text (sections 2.1-2.4)– this material has been

well covered in other literature. Section 2.5 provides a link with Paper 1, and thus should be included. 15. Following a discussion of periodic earthquakes (as arises from a spring-slider block sort of instability) on the heasl of Lotka-Volterra oscillators is a bit disingenuous, as the physics manifest in the PDEs are really quite different. 16. Line 307 – should be "Punchbowl" fault 17. Line 376 – update the 2020 reference. 18. Line 442 – again, should probably capitalize "Part 1" and will need to update the reference. 19. Line 492 – I disagree that cross diffusion terms from Onsager assumptions are the same or a "new" form of chemical wave at any scale, let alone at the "smallest scale. 20. Line 510 – the sentence is incomplete. 21. Line 532 – maybe reference a Turing paper for scholarly completeness? 22. Line 632 – should reference papers by Olsen and Olsen and Holcomb here 23. Line 685 – Finally! A promise of the application of cross-diffusional terms to earthquake triggering!! Oh wait, but first we need to hear about solitons!! 24. Line 721 – the paper is unfinished here – what figure is being referred to? Same in line 745 and line 749.

---

## Author Comment (AC1) · 19 May 2021

Anonymous Referee's #2 I had high expectations for this manuscript, having reviewed the "Part 1" paper of this two-part series. The premise for this current manuscript is that rogue fluid pressure waves can be quantified via cross coupling of off-diagonal diffusivity terms in reaction diffusion partial differential equations, and such waves are earthquake triggering. This is a companion paper to "Part 1" which described the general instability and coupling features. The paper offers a great high-end discussion of earthquake physics from the perspective of THMC processes across time and length scales and as such, it seems at the start of the paper that it offers a good overview and thus a good inclusion in the special issue. Placing an important instability such as earthquake triggering in the context of broader THMC instabilities is certainly an interesting precept for a paper. However I think the paper frequently veers off course, and during a first read every time I was met with a promise of discussing earthquake physics the paper lumbers off onto barely related topics (Lotka-Volterra chemical oscillators, solitons, dislocation theory, spinodal decomposition, and on and on.) Perhaps the paper would be more readable if these sections were omitted or at least shortened (we don't, for example, need a lengthy discussion on wave propagation, predator-prey problems, and spin- odal decomposition just to introduce cross terms in a diffusion matrix– this makes me think that part of this paper were in fact prepared originally as a textbook). Having al- ready reviewed "Part 1", I was really looking forward to the meat of this paper which is the mathematical workup to the cross diffusional rogue wave treatment in the context of earthquake instability, where much of this ancillary discussion would have already occurred in Part 1. It is perhaps too late to consider this option (i.e. placing the back- ground of THMC instabilities in paper 1, leaving paper 2 to focus on earthquakes). But that would be the ideal situation.

Reply: A focus on just the application of Part 1 to earthquakes was indeed the first aim of this sequel. Through discussions with interested parties it became, however, apparent that the concept of diffusional waves (even without considering cross-diffusion) was apparently not well established in the community. In order to alleviate this problem, the paper included an introduction to the topic. We agree, however, that this didactical material (although appreciated by the second reviewer) veers of course and is better placed in a review article or a textbook. The removal of the didactic material will indeed make more room for the important introduction of the rationale for the cross-diffusion terms and application to earthquakes and thus provide a much better angle for the paper. As the other reviewer appreciated the recap some of the material is now available

by replacing the old Appendix with this introduction into reaction-diffusion waves.

Anonymous Referee's #2 The linking of chemical oscillators to rogue waves as presented however is unfortu- nately ad hoc at best: although a summary of reaction-diffusion instabilities and related phenomena like Leisegang bands is perhaps an appropriate summary for a journal is- sue devoted to coupled THMC modeling, frankly the Leisegang phenomena and its related dynamics have little to do with cross diffusional terms and in particular with rogue waves. It seems the paper relies on the facts that both are "waves" to make a similarity argument, despite the inherent differences in the physics. The discussion on deformation bands is certainly appropriate but again the discussion loses robustness when linking to dislocation processes. It would do the paper well to focus on earth- quakes and cross diffusional phenomena as the title suggests and forgo all this extra, unnecessary, and frankly unrelated stuff.

Reply: This part is now removed

Anonymous Referee's #2 I am no expert in the propagation of electro-cardio waves and the associated instability, but the arguments presented in this paper would be better served if the so-called cross diffusional approach used to describe this (FitzHugh-Nagumo oscillator) were more fleshed out (section 4.1), rather than mentioned in passing. Time spent on analogy between the FitzHugh-Nagumo mechano-electro-chemical oscillator would do much to improve the paper and not the Volterra chemical wave dynamics which owe nothing to cross diffusional terms.

Reply: The removal of the didactic material has made space for just this discussion which is now added. We also no longer use in this paper the piecewise linearized analytical solution of the FitzHugh-Nagumo oscillator but instead the more relevant hydro-poro-mechanical oscillator which is described in details in a separate publication available and added as a preprint to this review (Sun et al. submitted).

Anonymous Referee's #2 By the time the paper starts actually discussing the earthquake instability, there are only a few pages left with only a disappointing arm-waving

level of treatment. I did not read the Discussion and Conclusion section. I think a paper linking seismically observed "tremors" as fluid-motion, rogue wave physics, the interesting rice-krispie earthquake-like experiments, discussions of com- paction waves, direct application of coarse-graining, and perhaps the FitzHugh- Nagumo oscillator (if at all applicable) would be a readable paper appropriate for this special issue. The rest just seems like unnecessary fluff. The appendices are unnec essary. I would recommend a rewrite along these lines.

Reply: This is a good suggestion, thank you. The paper focuses now on just the application to the earthquake problem.

Anonymous Referee's #2 Some additional comments: 1. Line 1/abstract - "Instabilities appears twice - change second occurrence to "source dominated source mechanisms". 2. Line 8 "These are here interpreted as a trigger. . ." 3. Line 13 "In this paper (Part 2) we investigate 4. Line 17 "Patterns in Our Planet" probably should be italicized 5. Line 23 – capitalize "Part 1" here and throughout the paper 6. Line 43 – don't capitalize "earth" 7. Line 63 – add coma after parenthesized term 8. Line 92 – does the Regenauer et al. 2020 paper refer to "Paper 1"? If so this reference needs to be updated. 9. Page 5, Figure Caption to Figure 2 – should provide a reference or two to the coarse graining discussed in the figure – (does this figure come from another reference? If so it should be treated accordingly. 10. Line 108 – should the linearized form of equation (1) contain a different notation for the "linearized" reaction rate expressed in equation 2? 11. Line 110 – should reference Table 1 here. 12. Line 170 – eliminate the first comma 13. Line 183 Eliminate the comma 14. The paper would lose nothing in range and scope of content if Figures 3-8 were eliminated, along with accompanying text (sections 2.1-2.4)– this material has been well covered in other literature. Section 2.5 provides a link with Paper 1, and thus should be included. 15. Following a discussion of periodic earthquakes (as arises from a spring-slider block sort of instability) on the Lotka-Volterra oscillators is a bit disingenuous, as the physics manifest in the PDEs are really quite different. 16. Line 307 – should be

"Punchbowl" fault 17. Line 376 – update the 2020 reference. 18. Line 442 – again, should probably capitalize "Part 1" and will need to update the reference. 19. Line 492 – I disagree that cross diffusion terms from Onsager assumptions are the same or a "new" form of chemical wave at any scale, let alone at the "smallest scale. 20. Line 510 – the sentence is incomplete. 21. 21. Line 532 – maybe reference a Turing paper for scholarly completeness? 22. 22. Line 632 – should reference papers by Olsen and Olsen and Holcomb here 23. Line 685 – Finally! A promise of the application of cross-diffusional terms to earthquake triggering!! Oh wait, but first we need to hear about solitons!! 23. 24. Line 721 – the paper is unfinished here – what figure is being referred to? Same in line 745 and line 749.

Reply: All of the minor suggestion have been adopted.

Please also note the supplement to this comment:
https://se.copernicus.org/preprints/se-2020-149/se-2020-149-AC1-supplement.pdf

---

## Author Response (AR1)

The paper discusses the THMC reaction-diffusion equations as underlying equations for the instabilities preceding earthquakes. I understand the efforts and troubles the authors met in preparing this paper: the topic is new for seismology and the theory grown in other fields had to be fit in the seismological frame. I admit that the paper results very difficult to follow in some parts. From one side, some parts are didactical (e.g. 2.1, 2.3), others are technical (2.2), other simply colloquial (section 4): and this dichotomy is more evident because the latter concerns the main topic of the paper.

**Reply:**

We would like to thank the referee for the frank assessment of the paper. We realise that the tightrope walk between trying to make the concepts clear in a didactical manner and providing deep insight into the newly proposed theory is not working. The anonymous referee commented on the same point and we have now removed some of the didactical content and moved every explanatory information into the Appendix.

**Angelo De Santis (Referee)**

Anyway, I found the paper very interesting and plenty of fresh indications for earthquake physics, with the presence of such as cross-diffusion waves and stationary waves. Nevertheless, I have two main concerns. The first regards the fact that the general theory overcomes the application to earthquakes very much. In this aspect the paper should be more balanced, in view of the fact that is dedicated to the application to earthquakes.

**Reply:**

This is true. The paper is now revised entirely with a focus on earthquakes and the discussion of the Liesegang patterns has been removed. The examples listed are now strictly only related to possible earthquake source mechanisms.

In the revised paper we have now provided clear mathematical proof that for the nucleation of an earthquake the cross-diffusion terms are important. In the original version we have only mentioned in the text that our equation merges into the nonlinear Schroedinger equation for particular coefficients. In this version we have explicitly shown for which coefficients this is happening and how this then leads to the Peregrine solution which can create rogue waves.

We have also been able to substantiate and illustrate the analysis by a generalized reaction-cross-diffusion equation which has been thoroughly studied in an independent paper (Sun et al. submitted). The code and the formulation are available as preprints and we hope that this easy access to our work will promote progress in the field.

**Angelo De Santis (Referee)**

My second concern is that the stationary waves have never been clearly found and neither weakly detected by earthquake seismology (apart from presumed but disputed findings present in Russian literature). If this is true, the entire presented framework, although intriguing and compelling, will be relegated only as a curiosity within a theoretical corner without any real application. This could be the most negative aspect of this work. My suggestion is to ask the Authors to work on the paper in order to make a "moderate" (more than minor, less than major) revision in order to let some concepts and their relationships more clear.

**Reply:**

It is true that we have incorporated too much material in the first submission in order to provide an easy introduction to the theory which diluted the novel concepts presented. In the revised edition we have now only focused on what is new in our approach and have moved didactic material into the appendix.

As to the KoYa or KaY standing wave we could not find a theoretical derivation. The waves are relatively slow and therefore they could be diffusion waves, however, this is not clear from the available material. What is obvious is that the KaY standing wave is a slow sonic wave whereas the waves modelled in our work are entirely dissipative and have creeping velocity. In our experiments the wave propagates for instance at 1 mm/s (new Figure 5) while the KoYa wave is reported to be around 100 km/h. If the KaY/Koya waves are indeed diffusion waves we could model them by including the elastic response using the framework presented in Part 1. If this is successful the resulting KoYa wave could be interpreted in terms of the Peregrine soliton but we have not done this in the present framework.

**Angelo De Santis (Referee)**

In the following, I list also some more punctual indications.

Lines 1-10 Abstract. I found it very short with respect the many information given in the main text. I suggest to expand it in order to include something more. By the way, the same indications (with the due adaptions) can be given for the section 5 of "Discussion and conclusions". See also below.

Line 28. I suggest Crampin et al. 2013 instead of Crampin and Gao 2015, as it takes the problem with a more seismological point of view, proposing the solutions for four typical seismological conondrums.

Lines 28-29 (also 64). I suggest the term "self-affinity" instead of "self-similarity".

Line 29. Better "log frequency- log magnitude relationship"

Line 34. "standing waves": how different are they from the "KaY waves" (Yagodin, 2017), whose existence is highly disputed (e.g. Koronovsky et al. 2019)?

Line 52. "holistic way...": how far is it from the "geosystemics approach" (De Santis 2009, 2014)? Probably they both share the same foundations.

said "... to first order ..." but the equation (2) is of the second orderin space. Caption Figure 3, first line: please correct "sale" with "scale".

Table 1. It is not fully clear how the given values of the table have been estimated.

Line 237. You mention the "autowaves" but you provide only one reference (Antonioletti et al. 2017). Are there some more references?

Line 279. "... with some apparent regularity in their recurrence ...", in the sense that the regularity is only apparent: the 2004 Parkfield event skipped the previous apparent periodicity.

Line 280. Please remove one bracket after "1997"

Line 292. Better "quasi-periodicity in space"?

Line 511. The line is broken. In addition, the present sentence is not clear.

Lines 569-570. The sentence "This approach ..." is not clear

Lines 721, 745, 749, 757. There are "???" for some figures.

Line 760. Pag. 31. Where is "Step 4" in the main text that is mentioned in the Figure 11 caption?

Lines 779-789. I think this part should be better exposed, because concerns the predictability or not of the cross-diffusion (here called rogue) waves. The conclusions are not clear: what I understand is that we should "leave the (present) analytical assessment ... aside" and pass to "investigate numerical solutions". But the next section (lines 790-798) poses many limits to this step.

Lines 895 and next ones. Web links are mostly double. Please check them.

Line 974. Why is the title of this reference in capital letters?

De Santis A., Geosystemics, Proceedings 3rd IASME/WSEAS International Conference on Geology and Seismology (GES'09), Cambridge, 36-40, 2009.

De Santis A., Geosystemics, entropy and criticality of earthquakes: a vision of our planet and a key of access, in "Nonlinear phenomena in Complex Systems: from Nano to Macro Scale" ed. E. Stanley and D. Matrasulov, NATO Science for Peace and Security Series – C: Environmental Security, 2014.

Koronovsky, N.V., Zakharov, V.S. & Naimark, A.A. Short-Term Earthquake Prediction: Reality, Research Promise, or a Phantom Project?. Moscow Univ. Geol. Bull. 74, 333–341 (2019). https://doi.org/10.3103/S0145875219040057

Yagodin, A.P., International Center of the Earthquake Prediction, 2017. https://sites.google.com/site/earthquakepredict/r1
* * *
**Reply:**  We have addressed the minor points and adopted the suggestions. Our holistic approach has many elements that are discussed in De Santis (2014) but our approach only provides the mathematics, the simplified physics and the forward modelling engine at this stage. De Santis proposes to use information theory (the Shannon Entropy) and the physics of fractal dimension , phase space, degrees of freedom, information and entropy as a geosystemic tool to overcome the limitation of empirical earthquake statistics. Such a geosystemic approach  is indeed important for a data assimilation step, where we could try to derive the coefficients of our pde's from data. As such the approach fits hand in glove. Our holistic approach unfortunately only provides the theoretical basis at this stage. We therefore cannot yet call it geosystemic.

The reason for not going further  and restricting ourselves to provide an in-depth presentation of the proposed physics and the discovery of a plethora of unknown phenomena in earthquake physics (slow quasi-soliton waves, rogue waves)  is that they still need to be found in seismological record. We have only proven that these wave exist in rock analogues by laboratory experiments. We have, therefore, not yet made the important step to compare the theory with seismological data. This should be the next logical step.

**Waves as a trigger for multiscale, multiphysics Instabilities: Application to earthquakes" *by* KlausRegenauer-Lieb et al.**

**Anonymous Referee's #2**

I had high expectations for this manuscript, having reviewed the "Part 1" paper of thistwo-part series. The premise for this current manuscript is that rogue fluid pressure waves can be quantified via cross coupling of off-diagonal diffusivity terms in reactiondiffusion partial differential equations, and such waves are earthquake triggering. Thisis a companion paper to "Part 1" which described the general instability and couplingfeatures. The paper offers a great high-end discussion of earthquake physics from theperspective of THMC processes across time and length scales and as such, it seemsat the start of the paper that it offers a good overview and thus a good inclusion in the special issue. Placing an important instability such as earthquake triggering in the context of broader THMC instabilities is certainly an interesting precept for a paper.

However I think the paper frequently veers off course, and during a first read every time I was met with a promise of discussing earthquake physics the paper lumbers off onto barely related topics (Lotka-Volterra chemical oscillators, solitons, dislocationtheory, spinodal decomposition, and on and on.) Perhaps the paper would be more readable if these sections were omitted or at least shortened (we don't, for example,need a lengthy discussion on wave propagation, predator-prey problems, and spin- odal decomposition just to introduce cross terms in a diffusion matrix– this makes methink that part of this paper were in fact prepared originally as a textbook). Having al-ready reviewed "Part 1", I was really looking forward to the meat of this paper which isthe mathematical workup to the cross diffusional rogue wave treatment in the contextof earthquake instability, where much of this ancillary discussion would have alreadyoccurred in Part 1. It is perhaps too late to consider this option (i.e. placing the back-ground of THMC instabilities in paper 1, leaving paper 2 to focus on earthquakes). But that would be the ideal situation.

**Reply:**

A focus on just the application of Part 1 to earthquakes was indeed the first aim of this sequel. Through discussions with interested parties it became, however, apparent that the concept of reaction diffusion waves (even without considering cross-diffusion) was apparently not well established in the community. In order to alleviate this problem, the paper included an introduction to the topic. We agree, however, that this didactical material (although appreciated by the second reviewer) veers of course and is better placed in a review article or a textbook. The removal of the didactic material has indeed made more room for the important introduction of the rationale for the cross-diffusion terms and application to earthquakes and thus provide a much better angle for the paper. As the other reviewer appreciated the recap some of the material is now available by replacing the old Appendix with this introduction into reaction-diffusion waves.

**Anonymous Referee's #2**

The linking of chemical oscillators to rogue waves as presented however is unfortu-nately ad hoc at best: although a summary of reaction-diffusion instabilities and related phenomena like Leisegang bands is perhaps an appropriate summary for a journal is-sue devoted to coupled THMC modeling, frankly the Leisegang phenomena and its related dynamics have little to do with cross diffusional terms and in particular with rogue waves. It seems the paper relies on the facts that both are "waves" to make a similarity argument, despite the inherent differences in the physics. The discussion on deformation bands is certainly appropriate but again the discussion loses robustness when linking to dislocation processes. It would do the paper well to focus on earth-quakes and cross diffusional phenomena as the title suggests and forgo all this extra, unnecessary, and frankly unrelated stuff.

**Reply:**

This part is now removed

**Anonymous Referee's #2**

I am no expert in the propagation of electro-cardio waves and the associated instability, but the arguments presented in this paper would be better served if the so-called cross diffusional approach used to describe this (FitzHugh-Nagumo oscillator) were more fleshed out (section 4.1), rather than mentioned in passing. Time spent on analogy between the FitzHugh-Nagumo mechano-electro-chemical oscillator would do much to improve the paper and not the Volterra chemical wave dynamics which owe nothing to cross diffusional terms.

**Reply:**

The removal of the didactic material has made space for just this discussion which is now added. We also no longer use in this paper the piecewise linearized analytical solution of the FitzHugh-Nagumo oscillator but instead the more relevant hydro-poro-mechanical oscillator which is described in details in a separate publication available and added as a preprint to this review (Sun et al. submitted).

**Anonymous Referee's #2**

By the time the paper starts actually discussing the earthquake instability, there are only a few pages left with only a disappointing arm-waving level oftreatment. I did not read the Discussion and Conclusion section.

I think a paper linking seismically observed "tremors" as fluid-motion, rogue wave physics, the interesting rice-krispie earthquake-like experiments, discussions of compaction waves, direct application of coarse-graining, and perhaps the FitzHugh-Nagumo oscillator (if at all applicable) would be a readable paper appropriate for this special issue. The rest just seems like unnecessary fluff. The appendices are unnec essary. I would recommend a rewrite along these lines.

**Reply:**

This is a good suggestion, thank you. The paper focusses now on just the application to the earthquake problem. Instead of referring to published literature which establishes a link of our approach to the Nonlinear Schrödinger equation we have now included a full discussion and corrected the mistakes in the reaction terms listed in

Tsyganov, M. A., et al. (2003). "Quasisoliton Interaction of Pursuit-Evasion Waves in a Predator-Prey System." Physical Review Letters **91**(21): 218102.

**Anonymous Referee's #2**

Some additional comments:

1. Line 1/abstract - "Instabilities appears twice - change second occurrence to "source dominated source mechanisms".

2. Line 8 "These are here interpreted as a trigger..."
3. Line 13 "In this paper (Part 2) we investigate
4. Line 17 "Patterns in Our Planet"
probably should be italicized
5. Line 23 – capitalize "Part 1" here and throughout thepaper
6. Line 43 – don't capitalize "earth"
7. Line 63 – add coma after parenthesizedterm
8. Line 92 – does the Regenauer et al. 2020 paper refer to "Paper 1"? If so this reference needs to be updated.
9. Page 5, Figure Caption to Figure 2 – should provide a reference or two to the coarse graining discussed in the figure – (does this figure come from another reference? If so it should be treated accordingly.
10. Line 108 – should the linearized form of equation (1) contain a different notation for the "linearized" reaction rate expressed in equation 2?
11. Line 110 – should reference Table 1 here.
12. Line 170 – eliminate the first comma
13. Line 183 Eliminate the comma
14. The paper would lose nothing in range and scope of content if Figures 3-8 were eliminated, along with accompanying text (sections 2.1-2.4)– this material has been well covered in other literature. Section 2.5 provides a link with Paper 1, and thus should be included.
15. Following a discussion of periodic earthquakes (as arises from a spring-slider block sort of instability) on the Lotka-Volterra oscillators is a bit disingenuous, as the physics manifest in the PDEs are really quite different.
16. Line307 – should be "Punchbowl" fault
17. Line 376 – update the 2020 reference.
18. Line442 – again, should probably capitalize "Part 1" and will need to update the reference.
19. Line 492 – I disagree that cross diffusion terms from Onsager assumptions are the same or a "new" form of chemical wave at any scale, let alone at the "smallest scale.
20. Line 510 – the sentence is incomplete.
21. 21. Line 532 – maybe reference a Turingpaper for scholarly completeness?
22. 22. Line 632 – should reference papers by Olsenand Olsen and Holcomb here
23. Line 685 – Finally! A promise of the application of cross-diffusional terms to earthquake triggering!! Oh wait, but first we need to hear about solitons!!
23. 24. Line 721 – the paper is unfinished here – what figure is being referred to? Same in line 745 and line 749.

**Reply:**
All of the minor suggestion have been adopted.

---

## Referee Report (RR1)

Reviewer Report

Ms Number:    se-2020-149

Full Title:    Cross-DiffusionWaves resulting from multiscale, Multiphysics instabilities: Application to earthquakes

Authors:    Regenauer-Lieb et al
* * *
GENERAL COMMENTS

This contribution builds on the Part I companion paper and applies the theory developed there to develop a new generic approach for illuminating the role that coupled THMC processes may play in triggering earthquake rupture (shear) instability through the generation of spatially and temporally organized spikes in field variables such as pore fluid pressure. The wider applications go far beyond the case of earthquake triggering, as complex THMC coupling and feedback processes, operating on different characteristic length and time scales, are clearly evident in the multiscale self-organization and self-similarity seen in geological structures and in geophysical observations.  Since the approach followed is so fundamental, it is inevitable that the paper is rather generic in nature and that significant introductory/didactic material has to be presented.  Nonetheless, I feel that the authors have responded very effectively to the criticisms levelled in the first round of the review procedure and have successfully focused the paper down more specifically on application to earthquake instability.  I found the paper, along with Part I, a fascinating read, and am encouraged to see how the authors and their co-workers are exploring the exceedingly challenging topic of reaction-diffusion processes and Onsager-type cross-coupling between the corresponding thermodynamic force-flux relations.  The paper will be quite abstract and unfamiliar for many readers. However, it is based on sound physical principles,  is well explained, well-referenced and well-structured. It should be welcomed into the literature as providing a new and inspirational basis for quantitatively exploring the complexity of geological and geophysical reality. In my judgement, the manuscript is acceptable for publication, though I would suggest a few minor revisions for the sake of clarity and completeness, as detailed below. No further review is needed in my opinion.

DETAILED  SUGGESTIONS

A) Abstract, line 11.  "a rogue wave would appear as a sudden fluid pressure spike on the future fault plane".  This suggests that only faulting of intact rock is important or considered here, which is not the case.  Best reformulate "future fault plane" to make clear that such a spike is most likely to trigger unstable slip on a pre-existing (near-critically stressed) fault.  This might be worth emphasizing in the main body of the article also.

B) Lines 15-16. "Although the paper is formulated for earth sciences the approach constitutes a generic theory for any material and therefore lacks experimental evidence." I do not see how this follows. Surely, the more general a theory is the more likely it would be that someone has found evidence for expression of some aspect of it.  Perhaps drop the last 5 words?  The issue of experimental evidence/validation is dealt with later, after all.

C) Lines 38-39. "The particles are subject to a pressure exerted by the solid matrix and the pressure of the pore fluid". Does this mean that only the hydrostatic component of (effective) stress supported by the solid is considered in the analysis?  In general, deviatoric stresses would be accounted for too, of course, but if that is the case here, would it not be better to replace "pressure exerted by the solid matrix" with something like 'thermodynamic force exerted by the stress supported by the solid matrix'?

D) Lines 179-183. I fully agree with the statement that is made here.  However, I would advise adding a short qualifier making it clear that "initial" material and environmental heterogeneities at multiple scales, will tend to complicate the picture.

E) Lines 189-194.  Quite a number of papers have been published on the structure and slip/seismogenic behaviour of faults in carbonate rocks, in nature and experiment, based on (quantitative) considerations of crystal plastic, diffusive and "superplastic" deformation mechanisms – and considering/reporting decarbonation more qualitatively. The recent papers by the Durham (e.g. De Paola 2015) and Utrecht (Chen, Verberne) groups come to mind, for example, as well as the somewhat broader work in a more rate-and-state framework by Aharanov and Scholtz. These mostly focus on feedback between shear/frictional heating and the above deformation (or asperity contact) mechanisms, rather than coupled reaction-diffusion in the present sense. Perhaps it is worth mentioning that the present type of model is not the only way of explaining localized fault structures in carbonates, or indicating how the approaches differ.

F) Line 331.  What is meant here by "effective stress weakens"?  Are the authors referring to a decrease in the effective (mean?) stress supported, i.e. to some measure of strength?

G) Lines 434-435. "In geomechanical laboratory experiments the solid matrix is known to respond to external forces by a nonlinear reaction commonly expressed in a power law".  Presumably a rheological law (power law creep) can be treated in the same way as a reaction rate description in the present treatment.  Both are dissipative kinetic processes, of course, but it might be useful to clarify briefly whether both can be treated equivalently.

H) Equations 12.  It would be helpful to many readers, I think, if a couple of lines could be added here to clarify what physical/chemical  conditions the chosen parameters imply.

---

## Author Response (AR2)

Reviewer Report

Ms Number:    se-2020-149

Full Title:    Cross-DiffusionWaves resulting from multiscale, Multiphysics instabilities: Application to earthquakes

Authors:    Regenauer-Lieb et al
* * *
GENERAL COMMENTS

This contribution builds on the Part I companion paper and applies the theory developed there to develop a new generic approach for illuminating the role that coupled THMC processes may play in triggering earthquake rupture (shear) instability through the generation of spatially and temporally organized spikes in field variables such as pore fluid pressure. The wider applications go far beyond the case of earthquake triggering, as complex THMC coupling and feedback processes, operating on different characteristic length and time scales, are clearly evident in the multiscale self-organization and self-similarity seen in geological structures and in geophysical observations.  Since the approach followed is so fundamental, it is inevitable that the paper is rather generic in nature and that significant introductory/didactic material has to be presented. Nonetheless, I feel that the authors have responded very effectively to the criticisms levelled in the first round of the review procedure and have successfully focused the paper down more specifically on application to earthquake instability. I found the paper, along with Part I, a fascinating read, and am encouraged to see how the authors and their co-workers are exploring the exceedingly challenging topic of reaction-diffusion processes and Onsager-type cross-coupling between the corresponding thermodynamic force-flux relations. The paper will be quite abstract and unfamiliar for many readers. However, it is based on sound physical principles, is well explained, well-referenced and well-structured. It should be welcomed into the literature as providing a new and inspirational basis for quantitatively exploring the complexity of geological and geophysical reality. In my judgement, the manuscript is acceptable for publication, though I would suggest a few minor revisions for the sake of clarity and completeness, as detailed below. No further review is needed in my opinion.

REPLY: We value very much the helpful comments of the reviewer which facilitated a rare process. Innovations in such a challenging topic are normally very hard to read and the exciting discoveries made may be lost to the broader readership. The review facilitated a process where through the concise application to the NLS the information entropy of the paper has decreased, and the paper is in a much sharper logical form. At the same time relegating the explanatory notes to the supplementary material the paper  remains available to a much broader readership that is unfamiliar with the complex feedbacks of thermodynamic force-flux relationships. We would like to thank the reviewer for making the additional effort of fine tuning through the minor suggestion and appreciate the enormous effort of providing such a detailed positive criticism.

DETAILED  SUGGESTIONS
A) Abstract, line 11. "a rogue wave would appear as a sudden fluid pressure spike on the future fault plane". This suggests that only faulting of intact rock is important or considered here, which is not the case. Best reformulate "future fault plane" to make clear that such a spike is most likely to trigger unstable slip on a pre-existing (near-critically stressed) fault. This might be worth emphasizing in the main body of the article also.

REPLY: Now changed to "In the context of hydromechanical coupling, a rogue wave would appear as a sudden fluid pressure spike. This spike is likely to cause unstable slip on a pre-existing (near-critically stressed) fault acting as a trigger for the ultimate (shear) seismic moment release."

B) Lines 15-16. "Although the paper is formulated for earth sciences the approach constitutes a generic theory for any material and therefore lacks experimental evidence." I do not see how this follows. Surely, the more general a theory is the more likely it would be that someone has found evidence for expression of some aspect of it. Perhaps drop the last 5 words? The issue of experimental evidence/validation is dealt with later, after all.

Reply: The 5 words have been deleted

C) Lines 38-39. "The particles are subject to a pressure exerted by the solid matrix and the pressure of the pore fluid". Does this mean that only the hydrostatic component of (effective) stress supported by the solid is considered in the analysis? In general, deviatoric stresses would be accounted for too, of course, but if that is the case here, would it not be better to replace "pressure exerted by the solid matrix" with something like 'thermodynamic force exerted by the stress supported by the solid matrix.'

REPLY: Now changed as suggested

D)Lines 179-183. I fully agree with the statement that is made here. However, I would advise adding a short qualifier making it clear that "initial" material and environmental heterogeneities at multiple scales, will tend to complicate the picture.

Reply: That is quite true. The qualifier has been added.

D)Lines 189-194. Quite a number of papers have been published on the structure and slip/seismogenic behaviour of faults in carbonate rocks, in nature and experiment, based on (quantitative) considerations of crystal plastic, diffusive and "superplastic" deformation mechanisms – and considering/reporting decarbonation more qualitatively. The recent papers by the Durham (e.g. De Paola 2015) and Utrecht (Chen, Verberne) groups come to mind, for example, as well as the somewhat broader work in a more rate-and-state framework by Aharanov and Scholtz. These mostly focus on feedback between shear/frictional heating and the above deformation (or asperity contact) mechanisms, rather than coupled reaction-diffusion in the present sense. Perhaps it is worth mentioning that the present type of model is not the only way of explaining localized fault structures in carbonates, or indicating how the approaches differ.

Reply: We have added a qualifier at the end of the paragraph stating that "We would like to emphasise at this point that chemical dehydration reaction-diffusion processes are only a special case of THMC feedback. A number of other recently proposed feedback processes need to be investigated for completeness, e.g. shear heating and phase changes triggered on asperity contacts (Hayward et al. 2016, Aharonov and Scholz 2019)."

E) Line 331. What is meant here by "effective stress weakens"? Are the authors referring to a decrease in the effective (mean?) stress supported, i.e. to some measure of strength?

Reply: replaced by strength

F) Lines 434-435. "In geomechanical laboratory experiments the solid matrix is known to respond to external forces by a nonlinear reaction commonly expressed in a power law". Presumably a rheological law (power law creep) can be treated in the same way as a reaction rate description in the present treatment. Both are dissipative kinetic processes, of course, but it might be useful to clarify briefly whether both can be treated equivalently.

Reply: We have added "Both chemical and mechanical reaction source terms are dissipative kinetic processes, and our approach can lead to the same laboratory-derived results. The difference is that the laboratory laws can strictly only be extrapolated for the laboratory conditions, while the physics-based approach is more generic."

G) Equations 12. It would be helpful to many readers, I think, if a couple of lines could be added here to clarify what physical/chemical conditions the chosen parameters imply.

Reply: Setting self-diffusion to zero implies that nonlocal cross-diffusion processes are happening very fast and trigger large internal-fluxes between solid and fluid. Additionally, the cross-coupled fluxes are maximized in normative sense by setting the cross-diffusion coefficients to unity and opposite sign. In other words, we consider the extreme condition where the ratio of cross-diffusion over self-diffusion coefficients tends to infinity.